# Differential chloride homeostasis in the spinal dorsal horn locally shapes synaptic metaplasticity and modality-specific sensitization

Francesco Ferrini [1,2,3,4,6 ✉], Jimena Perez-Sanchez[2,4,6], Samuel Ferland [2,4], Louis-Etienne Lorenzo [2], Antoine G. Godin [2,3,4], Isabel Plasencia-Fernandez [2,4], Martin Cottet[2], Annie Castonguay[2], Feng Wang [2], Chiara Salio[1], Nicolas Doyon[2,5], Adalberto Merighi[1] & Yves De Koninck [2,3,4]

GABA$_A$/glycine-mediated neuronal inhibition critically depends on intracellular chloride (Cl$^-$) concentration which is mainly regulated by the K$^+$-Cl$^-$ co-transporter 2 (KCC2) in the adult central nervous system (CNS). KCC2 heterogeneity thus affects information processing across CNS areas. Here, we uncover a gradient in Cl$^-$ extrusion capacity across the superficial dorsal horn (SDH) of the spinal cord (laminae I-II: LI-LII), which remains concealed under low Cl$^-$ load. Under high Cl$^-$ load or heightened synaptic drive, lower Cl$^-$ extrusion is unveiled in LI, as expected from the gradient in KCC2 expression found across the SDH. Blocking TrkB receptors increases KCC2 in LI, pointing to differential constitutive TrkB activation across laminae. Higher Cl$^-$ lability in LI results in rapidly collapsing inhibition, and a form of activity-dependent synaptic plasticity expressed as a continuous facilitation of excitatory responses. The higher metaplasticity in LI as compared to LII differentially affects sensitization to thermal and mechanical input. Thus, inconspicuous heterogeneity of Cl$^-$ extrusion across laminae critically shapes plasticity for selective nociceptive modalities.

[1] Department of Veterinary Sciences, University of Turin, Turin, Italy. [2] CERVO Brain Research Centre, Québec, QC, Canada. [3] Department of Psychiatry and Neuroscience, Université Laval, Québec, QC, Canada. [4] Graduate program in Neuroscience, Université Laval, Québec, QC, Canada. [5] Department of Mathematics and Statistics, Université Laval, Québec, QC, Canada. [6] These authors contributed equally: Francesco Ferrini, Jimena Perez-Sanchez.
✉email: francesco.ferrini@unito.it

Synaptic inhibition is mainly mediated by γ-aminobutyric acid (GABA) and glycine through the activation of GABA$_A$ and glycine receptors, respectively[1,2]. These receptors show high permeability to chloride (Cl$^-$) and low permeability to bicarbonate (HCO$_3^-$)[3]. Therefore, the strength of inhibition critically depends upon the intracellular concentration of Cl$^-$ ([Cl$^-$]$_i$), which is maintained low in adult central neurons by the K$^+$-Cl$^-$ co-transporter KCC2[4,5]. Alterations of KCC2 function are expected to disrupt synaptic inhibition by weakening the electrochemical Cl$^-$ gradient. Indeed, reduced KCC2 function has been implicated in many neurological disorders, such as epilepsy[6], motor spasticity[7], schizophrenia[8], stress[9], as well as in addiction[10,11]. Impaired KCC2 activity has been also identified as a key mechanism underlying neuropathic pain[12–14]. Aberrant pain results from the ensuing disinhibition that unmasks low threshold inputs to pain-relay neurons[15–17].

Pathological decrease in KCC2 levels underlying pain hypersensitivity has been consistently associated with the release of brain-derived neurotrophic factors (BDNF) from both neurons[18] and microglia[12,19]. Yet, BDNF, through its cognate receptor TrkB, can also regulate KCC2 in normal and developing tissue[20,21]. Notably, physiological [Cl$^-$]$_i$ is not constant, but varies across different cell populations[22–25]. Inconspicuous changes in KCC2 expression may impair the stability of the Cl$^-$ gradient, affecting spatial and temporal summation of inhibitory inputs, thus unsettling the control of firing activity[26–28]. Therefore, local variations in BDNF-TrkB signaling, may shape the behavior of inhibitory neuronal circuits by tuning Cl$^-$ homeostasis, thus altering the strength of inhibitory transmission. This form of plasticity, also known as ionic plasticity[29], in turn, impacts on the potentiality for plasticity at excitatory synapses (metaplasticity)[30]. In this respect, growing evidence indicates that long-term potentiation (LTP), is facilitated in presence of KCC2 hypofunction[31,32]. Surprisingly, the level of KCC2 activity in nociceptive circuits remains largely unknown and, hence no predictions can be made on the functional impact of uneven Cl$^-$ homeostasis on nociceptive processing and its plasticity.

The superficial dorsal horn (SDH) is an important site for the processing of nociceptive information. It is heterogeneous, with the majority of thermoceptive peptidergic afferents terminating in its outer portion, while non-peptidergic mechanoceptive C-fiber afferents in its inner portion. Lamina I (LI) constitutes the most important site of relay of nociceptive information to the brain, while lamina II (LII) consists in a highly packed layer of inhibitory and excitatory interneurons[33]. We determined the functional impact of uneven levels of KCC2 in nociceptive pathways, using a combination of in silico, in vitro and in vivo approaches.

Our results unmask a TrkB-dependent interlaminar gradient in KCC2 activity across the SDH, leading to greater lability of inhibition and activity-dependent ionic plasticity onto LI neurons. This results in very distinct forms of synaptic plasticity in the two laminae: a continuous facilitation of synaptic responses in LI neurons in contrast to a restrained plasticity in LII. The distinctive form of metaplasticity featured in LI, due to its heightened ionic plasticity, reveals a neural substrate underlying the propensity for catastrophic amplification and spread of hypersensitivity in a modality-specific manner.

## Results

**Heterogeneous Cl$^-$-extrusion capacity in the SDH**. Assessing Cl$^-$-extrusion capacity requires measurement under a Cl$^-$ load[5,19,34,35]. Thus, rat SDH neurons were recorded in whole-cell configuration, by applying a high Cl$^-$ load (29 mM) through the recording pipette[19,34,35] (Fig. 1a). Under these conditions, the theoretical value of $E_{GABA}$ according to the Goldman-Hodgkin-

Katz (GHK) equation is −37 mV. The calculated $E_{GABA}$ was experimentally confirmed by measuring $E_{GABA}$ from excised patch recordings in outside-out configuration (Fig. 1a). In this configuration, $E_{GABA}$ approached the theoretical value (−37.3 ± 5.5 mV, S.D., $n = 4$; Fig. 1b). Conversely, the experimentally obtained value of $E_{GABA}$ was about 10 mV more hyperpolarized in whole-cell configuration (−48.1 ± 8.0 mV, S.D., $n = 72$; Fig. 1b), revealing active Cl$^-$ extrusion. By applying the GHK equation, we estimated from $E_{GABA}$ the experimental value of [Cl$^-$]$_i$. The capacity to extrude Cl$^-$ was highly heterogeneous and the estimated intracellular concentration ranged from 5 to 30 mM (Fig. 1c), as reported in other brain areas[25,36]. Experimental conditions with high Cl$^-$ load were replicated in silico using a virtual neuronal model (Fig. 1d). Somatic [Cl$^-$]$_i$ had little dependence on both the extent of the dendritic tree and the dendritic extrusion capacity, since the rapid drop in [Cl$^-$]$_i$ in dendrites prevents Cl$^-$ extrusion (Fig. 1d). A Cl$^-$ gradient was observed at the tip of the pipette and extended for about 50 μm (Fig. 1d), confirming that the length of the modeled pipette was not critical beyond this range.

Bath-application of the specific NKCC1 antagonist bumetanide (10 μM) did not change $E_{GABA}$ (−43.7 ± 6.5 mV vs. −43.8 ± 6.5 mV, S.D., $n = 6$; Fig. 1e), whereas co-administration of the KCC2 and NKCC1 blocker furosemide (100 μM)[37] induced a depolarizing shift (−37.3 ± 7.1 mV, S.D.; one-way-repeated-measures-(RM)-ANOVA with Bonferroni post-hoc, $F = 15.5$, $P = 0.008$; Fig. 1e), indicating that Cl$^-$ extrusion occurred through KCC2. To assess whether KCC2 maintains Cl$^-$ homeostasis during activity-dependent Cl$^-$ load, repeated GABA puffs were applied in gramicidin-perforated configuration, to measure $E_{GABA}$ without altering the physiological [Cl$^-$]$_i$[38]. Repeated GABA applications produced a stable hyperpolarizing response, but not in the presence of furosemide, where the Cl$^-$ gradient rapidly collapsed, leading to a depolarizing response (Fig. 1f). We then estimated the Cl$^-$current associated with repeated GABA$_A$ activation in silico as predicted by our electrodiffusion neuronal model (Fig. 1g, h). The total Cl$^-$ conductance that best fit the extent of experimental Cl$^-$ accumulation was 2.5 nS. As expected, the same conductance in absence of Cl$^-$ extrusion led to an inversion of Cl$^-$ current polarity (Fig. 1g). The Cl$^-$ accumulation due to sustained GABA application induced a progressive change in $E_{Cl}$, to about 4 mV more depolarized, when Cl$^-$ transport was blocked (Fig. 1h). The requirement of a furosemide-sensitive Cl$^-$ transport to manage sudden increases in conductance was further confirmed experimentally by estimating the recovery of [Cl$^-$]$_i$ following a large conditioning GABA$_A$ current[39] (Supplementary Fig. 1). Therefore, although Cl$^-$-extrusion capacity is highly heterogeneous in the SDH, neurons appear to robustly maintain [Cl$^-$]$_i$ when challenged with a Cl$^-$ load, and this mainly depends on KCC2 activity.

**Differences in Cl$^-$-extrusion capacity between LI and II**. The large variability of Cl$^-$-extrusion capacity observed across the SDH suggests the existence of neuronal populations with different levels of Cl$^-$ transport capacity. To test for heterogeneity in Cl$^-$ extrusion, we sorted the recorded SDH neurons according to their laminar localization (LI and II) based on morphological criteria and distance from the dorsal white matter (LI neurons, 24.6 ± 2.5 μm; LII neurons, 83.6 ± 7.3 μm)[40]. The laminar localization was also confirmed by neurokinin receptor 1 (NK1) staining to detect cell bodies in LI[41], protein kinase Cγ (PKCγ) or isolectin B4 (IB4) staining to visualize LII[42,43], respectively (Fig. 2a).

$E_{GABA}$ in the presence of a Cl$^-$ load was significantly more depolarized in LI neurons (−45.9 ± 6.2 mV, S.D., $n = 50$; Fig. 2a) compared to LII (−53.2 ± 9.5 mV, S.D., $n = 22$; unpaired $t$-test,

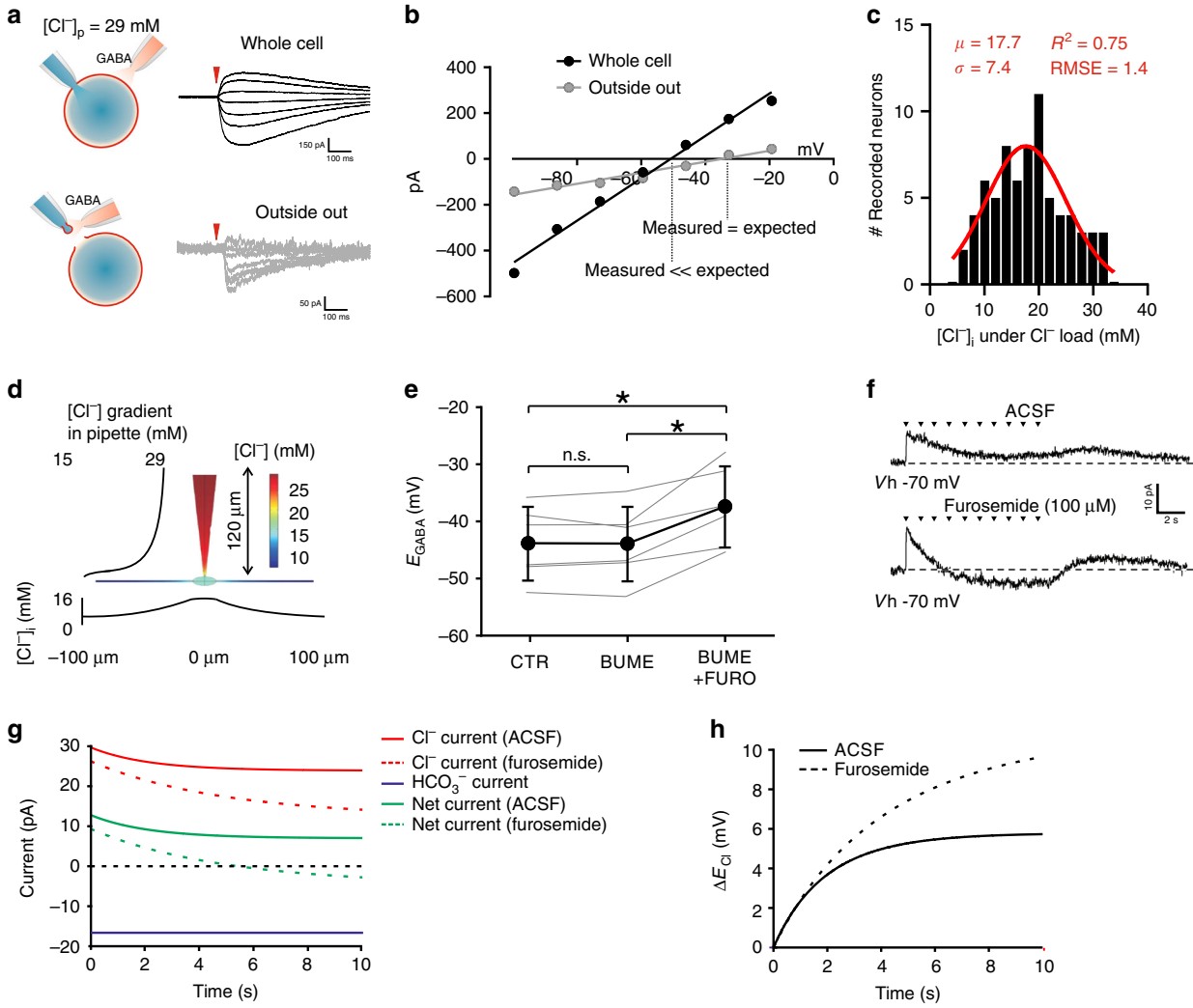

**Fig. 1 $Cl^-$-extrusion capacity in neurons from the superficial dorsal horn. a** Schematic representation of $Cl^-$ extrusion measurements in whole-cell configuration with a high-$Cl^-$ pipette solution (29 mM; *upper panel*) and on an excised patch of membrane (outside-out configuration; *lower panel*). GABA puffs are recorded at different holding potentials to determine $E_{GABA}$ (representative recordings, on the right). **b** GABA *I–V* curves from the neuron in **a** obtained in whole-cell configuration (*black circles*) and in outside-out configuration (*gray circles*). Experimental $E_{GABA}$ in whole-cell configuration (-48.1 ± 5.5 mV, $n = 72$) is more hyperpolarized than theoretical $E_{GABA}$ (-37 mV) or $E_{GABA}$ obtained in outside-out configuration (-37.3 ± 5.5 mV, $n = 4$). **c** Distribution of $[Cl^-]_i$, in SDH neurons estimated from experimentally measured $E_{GABA}$, under $Cl^-$ load (29 mM) ($n = 72$). **d** 3D neuronal modeling describing pipette imposed $Cl^-$ load, $Cl^-$ diffusion, as well as $Cl^-$ extrusion through KCC2 with color coding of $[Cl^-]_i$. **e** $E_{GABA}$ measured from SDH neurons ($n = 6$) in control (-43.7 ± 6.5 mV), in the presence of bumetanide (-43.8 ± 6.5 mV) and in the presence of bumetanide + furosemide (-37.3 ± 7.1 mV; *black*). Single neuron values, in gray. **f** Shift in $GABA_A$ current polarity in a SDH neuron upon sustained GABA administration (a train of 10 puffs-30 ms each, 1 Hz; *arrowheads*) at resting membrane potential in gramicidin-perforated patch before (*upper trace*) and after furosemide (100 μM; *lower trace*). Dashed line indicates the recording baseline. **g** Time course of in silico simulated $Cl^-$ (*red*), $HCO_3^-$ (*blue*) and net current (*green*) both in control (*solid lines*) and when KCC2 activity is blocked (*dashed lines*), as in presence of furosemide. Observe that the magnitude of the $Cl^-$ component responsible for $Cl^-$ accumulation and $E_{Cl}$ depolarization is larger than the net current measured. **h** Time course of in silico simulated $E_{Cl}$ depolarization both in control (*solid line*) and furosemide-like conditions (*dashed line*) taken from the simulations generating the currents shown in Fig. 1g. *μ* Gaussian fit mean, *σ* Gaussian fit S.D., RMSE root-mean-square deviation, CTR control, BUME bumetanide, FURO furosemide, n.s. not significant. Data are shown as mean ± S.D. *$P < 0.05$.

$t = 3.9$, $P < 0.001$; Fig. 2a). Irrespective of the laminar localization, $[Cl^-]_i$ values were broadly distributed, confirming a heterogeneous $Cl^-$-extrusion capacity across the SDH. However, the distribution of $[Cl^-]_i$ in LII was significantly different from that in LI (Kolmogorov–Smirnov test, $P < 0.01$ Fig. 2b), revealing weaker extrusion capacity in LI. Furosemide brought $E_{GABA}$ values close to the theoretical value (–37 mV) in both laminae (–39.5 ± 9.2 mV, S.D., $n = 9$; paired *t*-test, $t = 4.2$, $P = 0.003$, in LI; –42.1 ± 4.2 mV, $n = 4$, paired *t*-test, $t = 5.9$, $P = 0.098$ in LII; Fig. 2c), confirming KCC2 as the main $Cl^-$ extruder in mature SDH neurons.

In contrast, differences in $E_{GABA}$ between laminae were undetectable under near-physiological $[Cl^-]_i$, using either whole-cell recording with low $[Cl^-]$ pipette (9 mM) or gramicidin-perforated patch (Fig. 2d–f). When using a low $[Cl^-]$ pipette, experimental $E_{GABA}$ was again more negative than the theoretical value (–61 mV) or that obtained from excised patches (–66.6 ± 0.3 mV). However, no interlaminar differences were observed (LI: –73.6 ± 4.7 mV, S.D., $n = 8$; LII: –74.3 ± 2.7 mV, S.D., $n = 10$; unpaired *t*-test, $t = 0.4$, $P = 0.7$; Fig. 2e). Similarly, no significant interlaminar differences were observed when using gramicidin-perforated patch (LI: –77.3 ± 7.6 mV, S.D., $n = 9$;

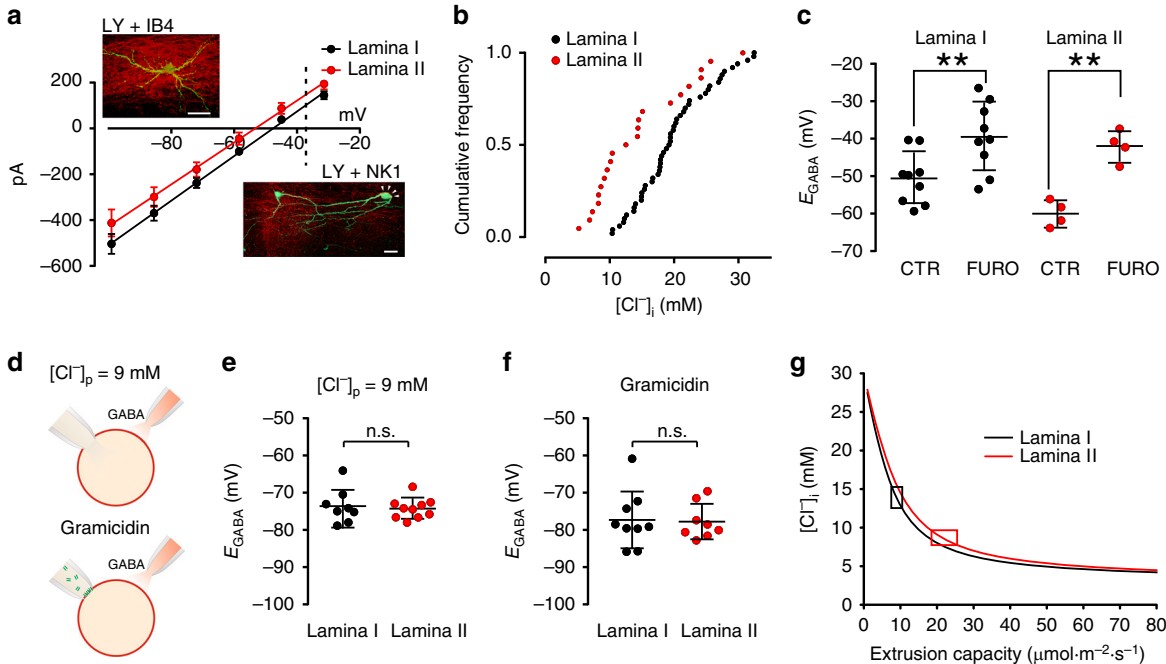

**Fig. 2 Heterogeneous Cl⁻-extrusion capacity across lamina I and II. a** Mean GABA I-V curves from LI ($E_{GABA}$ = −45.9 ± 6.2 mV, $n$ = 50; *black*) and LII ($E_{GABA}$ = (−53.2 ± 9.5 mV, $n$ = 22; *red*) neurons. *Dashed line* indicates the expected $E_{GABA}$ value according to the GHK equation. Insets show Lucifer-yellow-injected neurons in LI (*top*) and LII (*bottom*). Proper laminar localization is confirmed by NK1 and IB4 immunostaining (in *red*). Arrowheads show a NK1-positive recorded neuron. Scale bar = 20 μm. **b** Distribution of estimated [Cl⁻]$_i$ for LI ($n$ = 50; *black*) and LII ($n$ = 22; *red*). **c** $E_{GABA}$ measured in LI and LII in control (−50.6 ± 6.9 mV, $n$ = 9, *black* and −60.2 ± 3.5 mV, $n$ = 4, *red*, respectively) and after furosemide administration (−39.5 ± 9.2 mV, and −42.1 ± 4.2 mV, respectively). **d** Schematic representation of $E_{GABA}$ recording in whole-cell configuration with a low Cl⁻ pipette solution (9 mM; *upper panel*) and in gramicidin-perforated patch conditions (Cl⁻-impermeant channels in *green*; *lower panel*). **e** $E_{GABA}$ measurements between LI and LII neurons with a low Cl⁻ load (−73.6 ± 4.7 mV, $n$ = 8 and −74.3 ± 2.7 mV, $n$ = 10, respectively). **f** $E_{GABA}$ measured in LI and LII in the presence of physiological [Cl⁻]$_i$ in gramicidin-perforated patch clamp (−77.3 ± 7.6 mV, $n$ = 9 and −77.8 ± 4.8 mV, $n$ = 8, respectively). **g** Relative change of [Cl⁻]$_i$ as a function of KCC2 extrusion capacity in LI and LII. Cl⁻ diffusion simulations were taken from the neuronal model in Fig. 1d, considering LI and LII neuronal geometries and somatic [Cl⁻]$_i$. Boxes represent the range of extrusion capacities consistent with electrophysiology measurements. CTR control, FURO furosemide; n.s. not significant. Data are shown as mean ± S.D. *$P$ < 0.05, **$P$ < 0.01.

LII: −77.8 ± 4.8 mV, S.D., $n$ = 8; unpaired $t$-test, $t$ = 0.1, $P$ = 0.9; Fig. 2f). Thus, KCC2-dependent differences in Cl⁻-extrusion capacity were experimentally detectable only when a Cl⁻ load was applied. The Cl⁻-extrusion capacity that best fit the experimental data in our neuronal model was about 25 μmol m⁻² s⁻¹ in LII and 10 μmol m⁻² s⁻¹ in LI (Fig. 2g). Thus, a Cl⁻-extrusion capacity twice as high in LII compared to LI may explain the observed interlaminar differences in Cl⁻ homeostasis.

**Impact of uneven Cl⁻ extrusion on dynamic Cl⁻ accumulation.** A weak Cl⁻-extrusion capacity has little effect on the strength of inhibition under low Cl⁻ load, but may have dramatic consequences when Cl⁻ homeostasis is challenged by activity[9,34]. Thus, we performed intracellular Cl⁻ imaging in spinal cord slices under different synaptic drive (Fig. 3a). When synaptic activity in the SDH was increased by applying the TRPV1 agonist capsaicin[44] (2 μM; Fig. 3b and Supplementary Figs. 2 and 3), a slope in [Cl⁻]$_i$ was detected with higher values in LI and progressively lower toward LII (slope = −0.027 mM μm⁻¹; $F$ = 4.8, $P$ = 0.03; Fig. 3c). Conversely, in presence of a cocktail of inhibitors to block excitatory and inhibitory synaptic activity, [Cl⁻]$_i$ was not significantly correlated with the distance from the dorsal border (slope −0.007 mM μm⁻¹; $F$ = 0.4, $P$ = 0.6; Fig. 3c). The linear fittings in the two experimental conditions were significantly different ($F$ = 11.3, $P$ = 0.009). Thus, constitutive differences in Cl⁻-extrusion capacities are undetectable by Cl⁻ imaging in silent networks[24], but can be observed when synaptic activity is heightened. We explored the impact of uneven Cl⁻

extrusion capacities on $E_{Cl}$ under different scenarios of synaptic activity in silico (Fig. 3d). The model predicted that, while under resting conditions the interlaminar difference in [Cl⁻]$_i$ (Δ[Cl⁻]$_i$) is minimal, Δ[Cl⁻]$_i$ increases with synaptic activity, becoming noticeable at baseline conditions reported in vivo[45]. The increase is more dramatic when the frequency of excitatory events overcomes that of inhibitory events[5,26], as observed following capsaicin administration (Fig. 3b and Supplementary Figs. 2 and 3). Cl⁻ accumulation under increased excitatory activity is due to membrane depolarization caused by both excitatory activity and spiking (Supplementary Fig. 2b, c), which increases Cl⁻ driving force[5,46]. Moreover, the increase of [K⁺]$_o$ during high-frequency spiking activity further promotes Cl⁻ accumulation by dampening Cl⁻ transport via KCC2[26]. Our simulations suggest that the differences in Cl⁻ transport may lead to an interlaminar difference of 3–4 mM [Cl⁻]$_i$, yielding a 10–15 mV shift in $E_{GABA}$, when activity is increased (Supplementary Fig. 2). Altogether, these data indicate that LI is prone to activity-dependent Cl⁻ accumulation.

**Gradient of KCC2 expression in SDH.** To determine whether the interlaminar difference in Cl⁻ homeostasis reflected differential KCC2 expression, we measured the distribution of the transporter by immunohistochemistry. KCC2 labeling intensity appeared to gradually increase from the utmost dorsal horn border to deeper laminae (Fig. 4a). As described in the adult rat[42], we used calcitonin gene-related peptide (CGRP) and IB4 to delineate SDH laminae (Fig. 4b). To avoid biased quantification due to differences in the laminar size between sections, KCC2

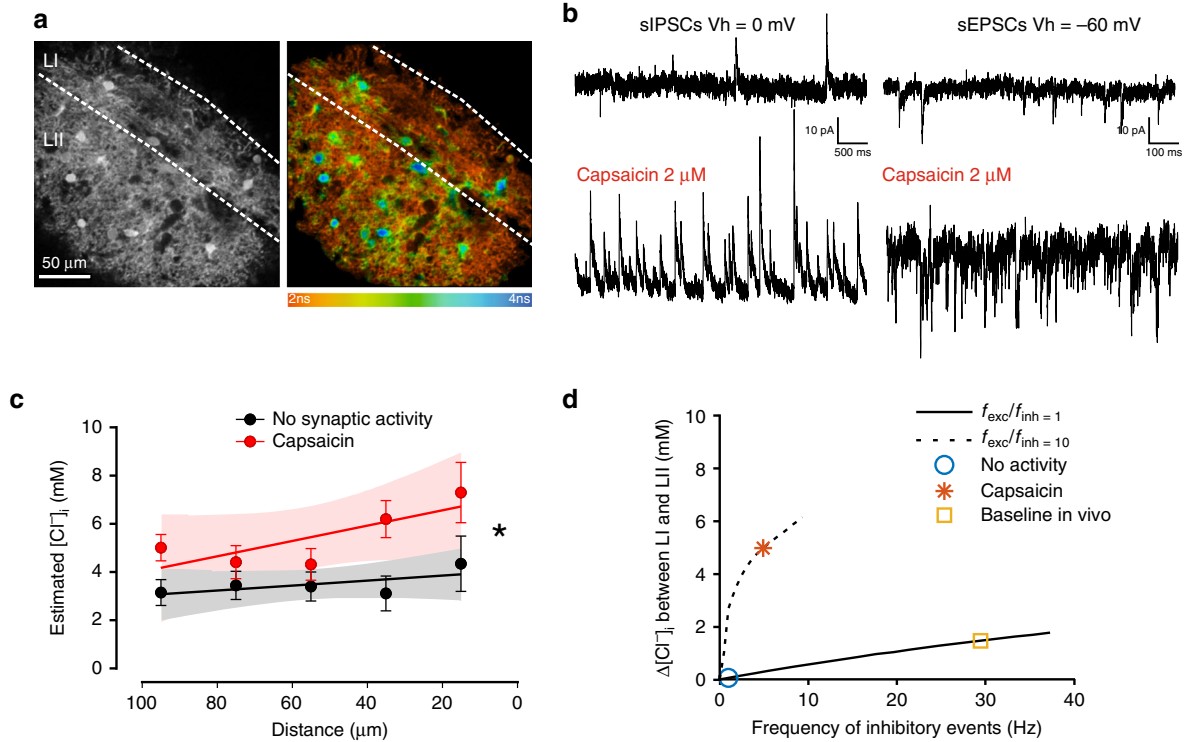

**Fig. 3 Activity-dependent [Cl$^-$]$_i$ gradient in the superficial dorsal horn. a** MQAE fluorescence in the SDH of a transverse spinal cord slice (*left*) and pseudocolor image (*right*) show heterogeneous Cl$^-$ concentrations across the SDH. **b** Representative patch clamp recordings from SDH neurons illustrate the strong increase in the frequency of spontaneous inhibitory (left) and excitatory (right) post-synaptic currents upon capsaicin administration (2 μM). **c** Estimated [Cl$^-$]$_i$ from MQAE fluorescence lifetime of SDH neurons is not correlated to the distance from the dorsal white matter in the presence of TTX (1 μM), bicuculline (20 μM), strychnine (1 μM), CNQX (10 μM), AP5 (40 μM) to block synaptic activity (*black*; slope −0.007 mM μm$^{-1}$; *n* cells per point = 20–43, total = 179 cells; F = 0.4, P = 0.6), while the correlation is significant in presence of capsaicin to enhance synaptic activity (*red*; slope = –0.027 mM μm$^{-1}$; *n* cells per point = 24–49, total = 209 cells; F = 4.8, P = 0.03). Two-way-ANOVA, F$_{treatment}$ = 17.2, P < 0.001. [Cl$^-$]$_i$ values in SDH neurons are binned every 20 μm. **d** In silico model of the differential impact of somatic synaptic input on [Cl$^-$]$_i$ (Δ[Cl$^-$]$_i$) between LI and LII. A first scenario considers the frequency of excitatory events equal to the frequency of inhibitory events (*solid line*), which approximates baseline and in vivo conditions, whereas a second scenario considers a frequency of excitatory events ten times higher than the frequency of inhibitory events (*dashed line*), similar to what we observed experimentally in slices under capsaicin application. Simulations are based on extrusion capacities estimated in Fig. 2g. LI Lamina I, LII lamina II, IPSCs inhibitory post-synaptic currents, EPSCs excitatory post-synaptic currents. Data are shown as mean ± S.E.M. *P < 0.05.

profile intensity was expressed as a function of the distance from the IB4 barycentre (Fig. 4c). Lower KCC2 expression was observed at the CGRP fluorescence peak (roughly corresponding to LI) while it progressively increased in LII toward the IB4 barycentre in both the rat (*n* = 8; Fig. 4c) and mouse (*n* = 8; Supplementary Fig. 4) SDH. To ensure that the observed inter-laminar KCC2 gradient was not due to differences in neuronal densities, we also directly measured KCC2 intensity on the membrane of randomly selected SDH neurons. The result confirmed a gradient in plasmalemmal KCC2 protein expression across laminae (linear regression slope = 3.4 i.u. μm$^{-1}$, *n* = 6; F = 15.5, P = 0.008; Fig. 4d). The differential expression of KCC2 was further analyzed at the membrane level by two additional custom-made methods for quantifying the fluorescence intensity per pixel[47,48]. A semi-automated method (membrane analysis of sub-cellular profile intensity; MASC-π) was used to measure KCC2 pixel intensities across sub-cellular compartments of identified neurons, by plotting the fluorescence signal as a function of the distance from the neuronal membrane[47] (Fig. 4e). The membrane KCC2 intensity per pixel was significantly lower in LI than in LII (*n* = 8; paired *t*-test, t = 2.9, P = 0.02). A significant difference was also detected with a second, automated method (membrane analysis using global index; MAGI), which estimates the global membrane intensity in each lamina by subtracting the

intracellular intensity from the total intensity[49] (*n* = 8, paired *t*-test, t = 4.6, P = 0.002; Fig. 4f). Independently, all four quantification methods confirmed a differential expression of KCC2 in LI and LII.

Since TrkB signaling plays a role in the control of KCC2 expression under different pathological conditions[11,12,19,21] and BDNF is expressed in peptidergic afferent terminals ending in the outer portion of the SDH[50], we asked whether the differential expression of KCC2 was under TrkB control. Following a single intraperitoneal injection of the TrkB antagonist ANA-12 in rats (0.5 mg kg$^{-1}$, 4 h before sacrifice)[51], we found that KCC2 was increased in LI (Fig. 5a, b), confirmed by MASC-π quantification (*n* = 6 per group; unpaired *t*-test, t = 2.6, P = 0.02; Fig. 5c) and MAGI quantification (unpaired one-tailed *t*-test, t = 1.9, P = 0.04; Fig. 5d). In contrast, no changes occurred in LII (unpaired one-tailed t = 0.6, P = 0.28; Fig. 5d). ANA-12 differentially affected KCC2 expression according to the laminar localization (two-way RM-ANOVA *n* = 6, F$_{interaction}$ = 8.7, P = 0.014; Fig. 5d). To investigate whether the effect can arise from a differential expression of the TrkB receptor in the SDH, we analyzed the ultrastructural localization of TrkB receptors in identified LI and LII neurons using fluoronanogold immunolabeling (Fig. 5e, f). As previously shown[50], TrkB is highly expressed across the SDH (Fig. 5e). However, we found a higher level of aggregation of

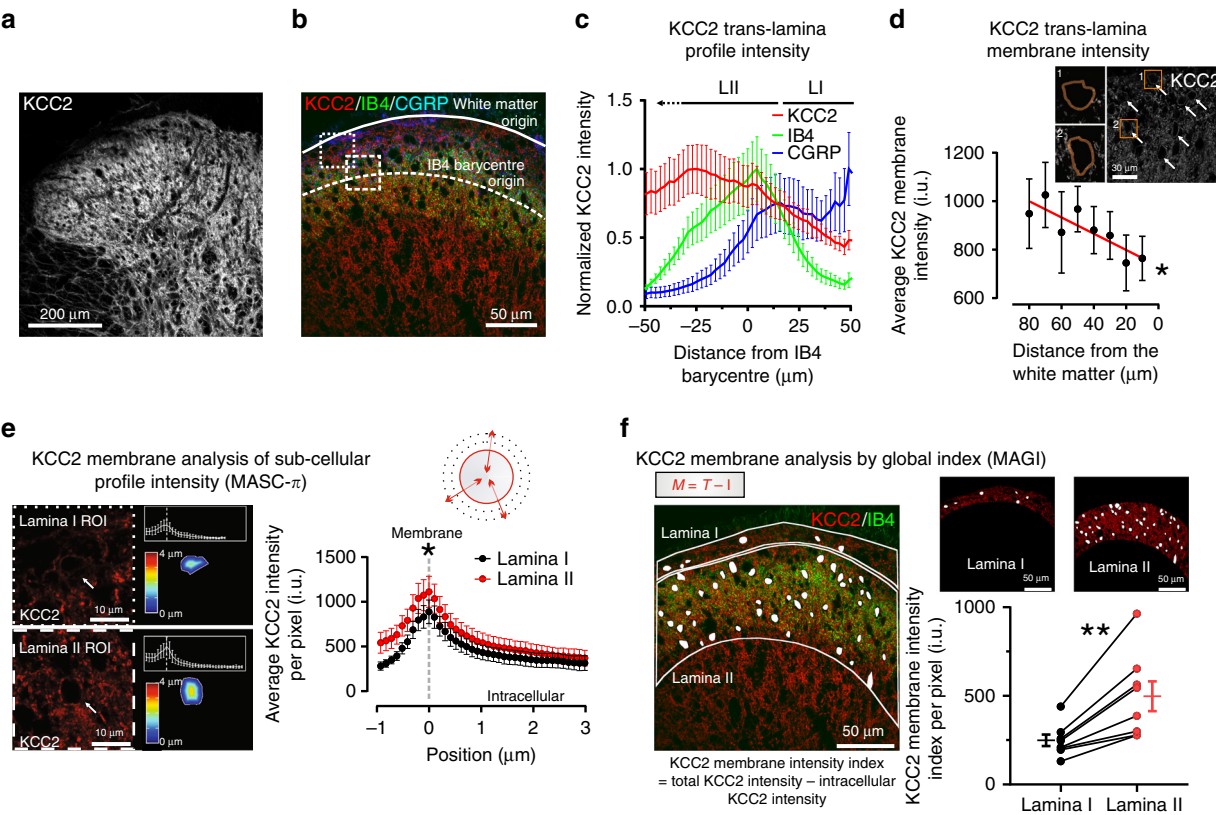

**Fig. 4 Interlaminar gradient of KCC2 in the superficial dorsal horn. a** Confocal image of KCC2 staining in the rat SDH. **b** CGRP (*blue*), IB4 (*green*), and KCC2 (*red*) staining in the rat SDH (dorsal white matter border, *solid line*; barycentric origin of IB4, *dashed line*). Boxed areas represent regions of interest (ROIs) surrounding LI (*small dash*) and LII neurons (*big dash*). **c** CGRP/IB4/KCC2 mean fluorescence intensities in the SDH as a function of the position towards the IB4 barycentric origin ($n = 8$ rats). **d** Membrane KCC2 intensity (*arrows*) as a function of distance from the dorsal white matter ($n = 6$). Insets, ROIs of neurons located superficially (1) and deeper (2) in the SDH (*boxed areas*). The slope of KCC2 intensity across SDH is significantly different from 0 ($F = 15.5$, $P = 0.008$); one-way-RM-ANOVA, $n = 6$, $F = 18.5$, $P < 0.001$. **e** Membrane analysis of sub-cellular profile intensities (MASC-$\pi$). On the left, KCC2-expressing neurons (*arrows*) in LI and LII from ROIs in **b**. The color-coded distance map illustrate the shift in KCC2 intensity with the distance to the membrane profile. The graphs in the insets quantify the KCC2 intensity as a function of distance to the membrane profile. On the right, pooled KCC2 values per rat as a function of distance to the membrane profile for all LI ($n = 8$) and LII ($n = 8$). The pictogram on the top right is a schematic representation of the method (**f**). Membrane analysis by global index (MAGI). On the left, ROIs containing LI and LII, delineated based on IB4 staining (in *green*). The global intensity index of the membrane KCC2 intensity per pixel (M, *red*) in each lamina is measured by automated subtraction of intracellular intensity (I, *white*) from the total intensity (T). *Top right*, demonstrative LI and II ROIs. *Bottom right*, quantification of membrane KCC2 signal in LI and LII by global intensity index. i.u. intensity units, CGRP calcitonin gene-related peptide, IB4 isolectin B4. Data are shown as mean ± S.E.M. *$P < 0.05$, **$P < 0.01$.

full-length TrkB-associated particles at the plasma membrane in LI cell bodies, compared to LII (Fig. 5f). The level of particle aggregation on the cell membrane was quantified by analyzing the distances of each fluoronanogold particle from every other particle (Fig. 5g). Examination of the distribution of interparticle distances in LI revealed a peak below 50 nm. This result indicates a non-random distribution of particle distances, suggestive of oligomerization. This peak was absent for LII (a smaller peak was observed > 80 nm). Separating particles using a cutoff distance of 65 nm revealed a significantly asymmetric distribution (Fisher's exact test, $P < 0.001$; Fig. 5g). The result suggests a higher level of receptor oligomerization in LI than LII. As receptor oligomerization is associated with a greater level of activity[52], the lower level of KCC2 expression in LI appears to result from higher level of TrkB activation in these neurons[53,54], consistent with our pharmacological results (Fig. 5a–d). Taken together these results suggest a tonic TrkB signaling in LI, causing differential KCC2 expression across laminae.

**Interlaminar differences in KCC2 affect ionic plasticity.** The heterogeneity in KCC2 expression and activity-dependent Cl⁻ accumulation may affect how neurons integrate inhibitory

inputs[5,46]. To investigate activity-dependent plasticity at inhibitory synapses, we focally stimulated inhibitory neurotransmission at holding potentials that lead to either a dominant Cl⁻ influx ($V_h = 0\,mV$) or a dominant $HCO_3^-$ outflux ($V_h = -90\,mV$) (Fig. 6a)[9,10,49]. During trains of stimulation, at sub-maximal frequency enough to challenge $E_{Cl}$, evoked IPSCs (eIPSCs) undergo an activity-dependent synaptic depression. However, the size of the depression is greater when eIPSCs are outwardly directed (involving a dominant Cl⁻ influx; Fig. 6a). This is due to the Cl⁻ accumulation occurring as a result of the barrage of GABA_A inputs. Thus, neurons with reduced Cl⁻-extrusion capacity should show a more dramatic activity-dependent eIPSC depression. Indeed, a more pronounced eIPSCs depression was observed in LI neurons ($n = 11$) at the holding potential of 0 mV than in LII ($n = 8$; $F = 30.2$, $P < 0.001$; Fig. 6b). In contrast, the course of eIPSC depression was not different at −90 mV when IPSCs are largely Cl⁻-independent ($F = 0.6$, $P = 0.6$; Fig. 6c). Since the $HCO_3^-$ driving force does not collapse under a GABA_A drive[9,55,56], synaptic depression measured at −90 mV reflects Cl⁻-independent GABA_A-current depression (e.g., desensitization, or presynaptic depression). Subtracting the eIPSCs depression measured at 0 mV from that at −90 mV allowed isolating the

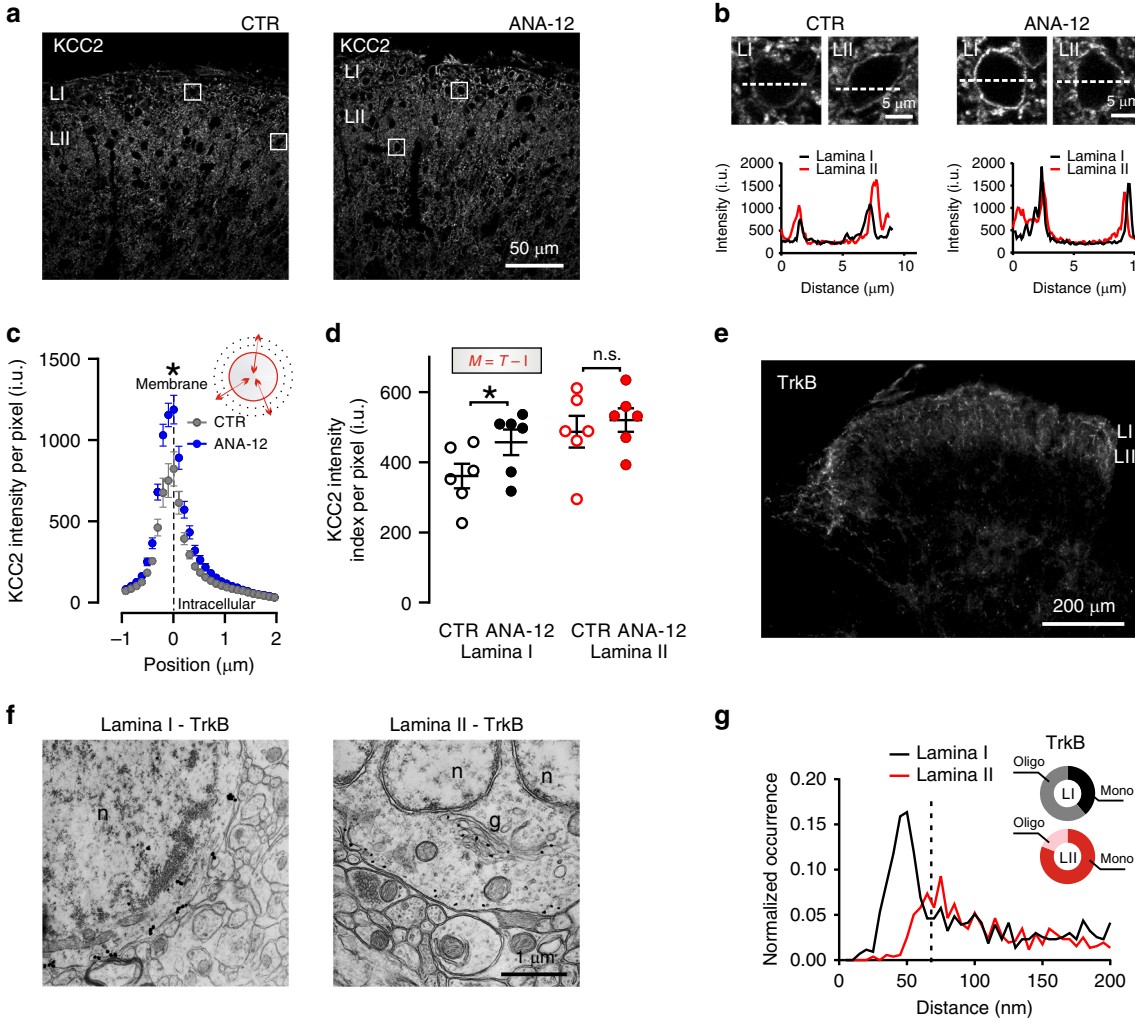

**Fig. 5 TrkB signaling impacts the interlaminar KCC2 gradient. a** Representative confocal images of KCC2 expression in SDH in control and ANA-12 treated rats. **b** Enlargements from the boxes in **a** show membrane KCC2 intensity in representative laminae I and II cells and correlative profile plots of pixel intensities across the cell bodies in a 10 μm segment (*dashed lines*). **c** KCC2 membrane profile intensity per pixel (see Fig. 4e) measured in LI (0–20 μm from the white matter) of vehicle- ($n = 6$) and ANA-12-treated rats ($n = 6$). **d** Global intensity index ($M = T – I$) in LI (0–20 μm) and LII (20–80 μm from the white matter) in control and ANA-12 treated rats. **e** Strong staining of full-length TrkB immunofluorescence in the rat dorsal horn obtained with FluoroNanogoldTM probe. **f** Ultrastructural labeling showing the somatic localization of full-length TrkB receptor in LI–II neurons. Labeling is mostly localized at the membrane level, although gold-intensified particles can also be found in the Golgi apparatus and in the rough endoplasmatic reticulum. **g** Frequency distribution of the interparticle distances in LI (*black*) and LII (*red*). Note the different position and size of the distribution peaks indicating a higher degree of particle proximity in LI. Dashed line indicates the cutoff interparticle distance to discriminate between monomers and oligomers (65 nm). Pie graphs in the inset illustrate the proportion of paired gold particles whose distance is shorter (*gray/pink*: putative oligomers) or larger (*black/red*: putative monomers) than 65 nm in LI (61%, $n = 506$ particles and 39%, $n = 320$ particles, respectively) and in LII (19%, $n = 133$ particles and 81%, $n = 568$ particles, respectively). CTR control, ANA-12 N-[2-[[(Hexahydro-2-oxo-1H-azepin-3-yl)amino]carbonyl]phenyl]benzo[b]thiophene-2-carboxamide, i.u. intensity units, n.s. not significant. Data are shown as mean ± S.E.M. *$P < 0.05$, **$P < 0.01$.

component specifically due to Cl⁻ accumulation. The subtraction between 0 and –90 mV depression in LI was greater than in LII (two-way ANOVA, $F_{interaction} = 4.1$, $P = 0.03$, post-hoc Bonferroni, $P = 0.003$; Fig. 6d). In contrast, no differences in eIPSCs collapse were observed when slices were pre-incubated with the TrkB antagonist ANA-12 (1 μM[51], post-hoc Bonferroni, $P > 0.9$) nor with the specific KCC2 enhancer CLP257 (5 μM;[10,49] post-hoc Bonferroni, $P > 0.9$; Fig. 6d and Supplementary Fig. 5). Thus, both reversing TrkB-dependent KCC2 downregulation or directly enhancing KCC2 activity, levels off interlaminar differences in inhibitory synaptic plasticity.

A corollary to these findings is that enhancing Cl⁻ influx may fail to improve inhibition when Cl⁻ extrusion is weak. To test this, we pharmacologically increased the GABA$_A$ conductance by

applying benzodiazepines[57,58]. We compared the depression of monosynaptic eIPSC amplitude in LI and LII upon repetitive stimulation before and after bath-application of the benzodiazepine diazepam (1 μM) at 0 mV where Cl⁻ influx dominates (Fig. 6e, f). Diazepam produced a significantly greater activity-dependent depression of eIPSCs, particularly in LI neurons, yielding a nearly complete collapse of the current amplitude by the end of the train (Fig. 6e, f). The net effect of the benzodiazepine on total charge transfer by the end of the stimulus train was negligible in LI, while it produced a significant increase in LII (paired t-test, $n = 8$, $t = 4.6$, $P = 0.002$; Fig. 6g). In conclusion, ionic plasticity is differentially modulated in the SDH by uneven Cl⁻-extrusion capacity, which differentially affects the robustness of inhibitory transmission. Weak Cl⁻ extrusion in LI

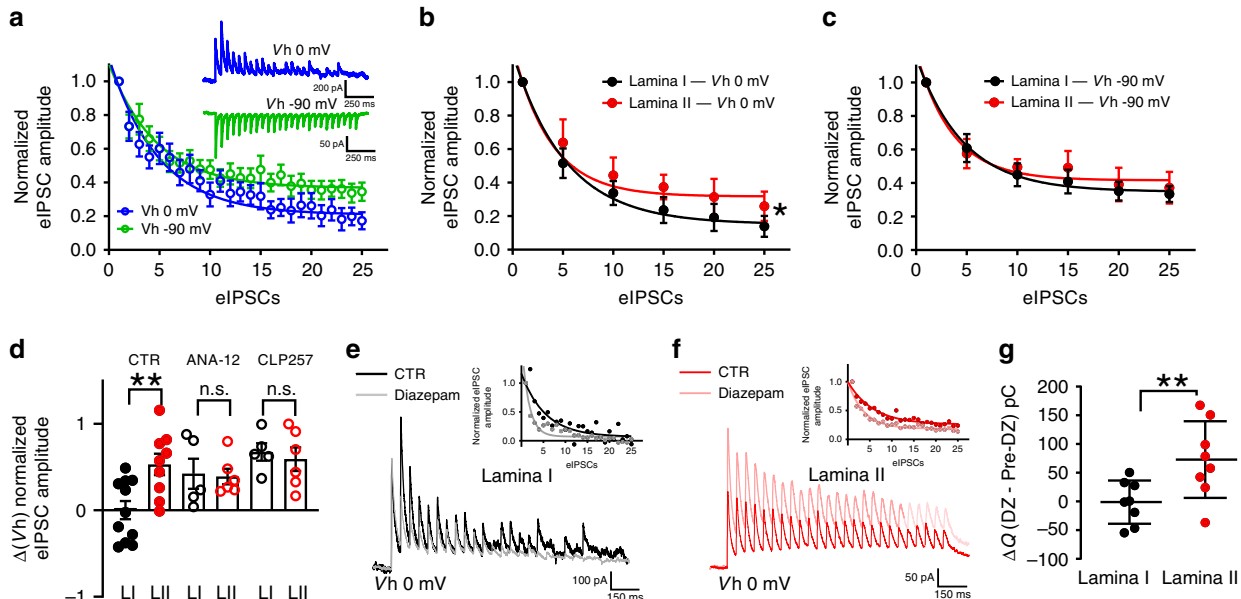

**Fig. 6 Activity-dependent collapse of inhibition in spinal lamina I. a** Depression of eIPSC amplitudes at 0 mV (*blue*) and −90 mV (*green*). Each data point represents the mean of 19 SDH neurons. Amplitude values are normalized to the first eIPSC. The course of eIPSC depression was fit by a one-phase decay exponential function. Notice the larger depression observed at 0 mV due to Cl⁻ accumulation. Insets show representative traces (average of ten repetitions) of repetitive inhibitory stimulation (20 Hz) at 0 mV (*top*) and −90 mV (*bottom*). Stimulus artefacts have been canceled. **b** eIPSC depression recorded at 0 mV is greater in LI ($n = 11$; *black*) than II ($n = 8$; *red*) neurons ($F = 30.2$, $P = 0.0002$). **c** No differences in eIPSC depression were observed at −90 mV between LI and II neurons. ($F = 0.6$, $P = 0.6$). **d** Difference between the rate of depression at $V_h$ 0 and −90 mV at the end of the repetitive stimulation in LI and LII in control condition or after pre-treatment with ANA-12 (1 μM) and CLP257 (5 μM). **e** LI recording (average of ten repetitions) of repetitive inhibitory stimulation (20 Hz) at 0 mV in control (*black*) and in presence of diazepam (1 μM, *gray*). The inset illustrates the variation of eIPSC relative amplitude across the stimulation. **f** LII recording (average of ten repetitions) of repetitive inhibitory stimulation (20 Hz) at 0 mV in control (*red*) and in the presence of diazepam (1 μM, *pink*). The inset illustrates the variation of eIPSC relative amplitude across the stimulation. **g** Difference between charge transfer (ΔQ) during the train of eIPSCs after and before diazepam in LI ($n = 8$) and LII ($n = 8$) neurons. CTR control, ANA-12 N-[2-[[(Hexahydro-2-oxo-1H-azepin-3-yl)amino]carbonyl]phenyl]benzo[b]thiophene-2-carboxamide, eIPSC evoked inhibitory post-synaptic currents, DZ diazepam, n.s. not significant. Data are shown as mean ± S.E.M. **$P < 0.01$.

leads to labile inhibition, prone to failure, which cannot be efficiently compensated for by enhancing Cl⁻ influx through GABA_A receptors[49].

**Higher ionic plasticity in LI yields distinctive metaplasticity.** Failure of inhibition upon sustained input is expected to shape activity-dependent excitatory synaptic strengthening[31]. To test whether differential Cl⁻ homeostasis across the SDH affects excitatory synaptic plasticity[31,59], we recorded field post-synaptic potentials (fPSP) evoked in mouse spinal cord explants at different distances from the dorsal surface[60]. A canonical protocol for producing long-term facilitation (LTF) was applied by electrically stimulating the dorsal roots at low frequency (2 Hz[60,61]; Fig. 7a). The relationship between input from primary afferents and the output in SDH neurons is highly nonlinear and generates high-frequency bursts in interneurons[62], challenging Cl⁻ homeostasis (Fig. 3d). We found that synaptic potentiation grew continuously without stabilization in superficial recordings (runaway LTF; $n = 8$), while it was restrained and stabilized in deeper recordings ($n = 5$, Two-way-RM-ANOVA, $F_{interaction} = 4.9$, $P < 0.001$; Fig. 7b). This LTF had grown to significantly larger values in superficial vs. deeper recordings by 150 min (two-way-RM-ANOVA, $F_{interaction} = 6.8$, $P = 0.02$; Fig. 7c). We found a significant correlation between the slope of individual LTF growth and the depth of the recording electrode ($R^2 = 0.34$, $P = 0.04$; Fig. 7d).

To test whether the runaway form of LTF observed in the superficial recordings was linked to weaker KCC2 expression, we

used the selective KCC2 antagonist VU0240551 (10 μM[15]) to block KCC2 function uniformly across laminae. Since KCC2 is not expressed in primary afferents[63], nor synaptic terminals in the SDH[13], VU0240551 is expected to exclusively affect Cl⁻ homeostasis at the post-synaptic level. We found that VU0240551 abolished the difference in synaptic plasticity across the SDH, by converting the restrained LTF in deeper recordings ($n = 5$) to runaway LTF, as in superficial recordings ($n = 5$; two-way-RM-ANOVA, $F_{interaction} = 0.6$, $P = 0.9$; Fig. 7e, g and Supplementary Fig. 6a). Similar results were also obtained by applying furosemide (100 μM) in explants continuously treated with bumetanide (10 μM; to isolate the effect on KCC2; Supplementary Fig. 7a, b). In addition, Cl⁻ loading of SDH neurons by prolonged activation of the Cl⁻ pump halorhodopsin (NpHR3.0) converted potentiation in deeper recordings into runaway LTF[59]. This procedure effectively equalized LTF in superficial and deep recordings (Supplementary Fig. 7c, d). Altogether, these results indicate that stronger Cl⁻ handling in deeper laminae is responsible for constraining facilitation.

As the differential expression of KCC2 across laminae relies on TrkB activation, we used the TrkB antagonist, ANA-12 (1 μM), to assess the contribution of the receptor to synaptic potentiation. ANA-12 treatment converted the LTF in superficial recordings ($n = 7$) to a restrained LTF not significantly different from that found in deeper recordings ($n = 6$; Two-way-RM-ANOVA, $F_{interaction} = 0.4$; $P = 0.9$; Fig. 7f, g and Supplementary Fig. 6b). Similarly, enhancing the activity of KCC2 with CLP257 (5 μM) stabilized synaptic plasticity across the SDH, thus abolishing the differences between superficial and deep recordings (two-

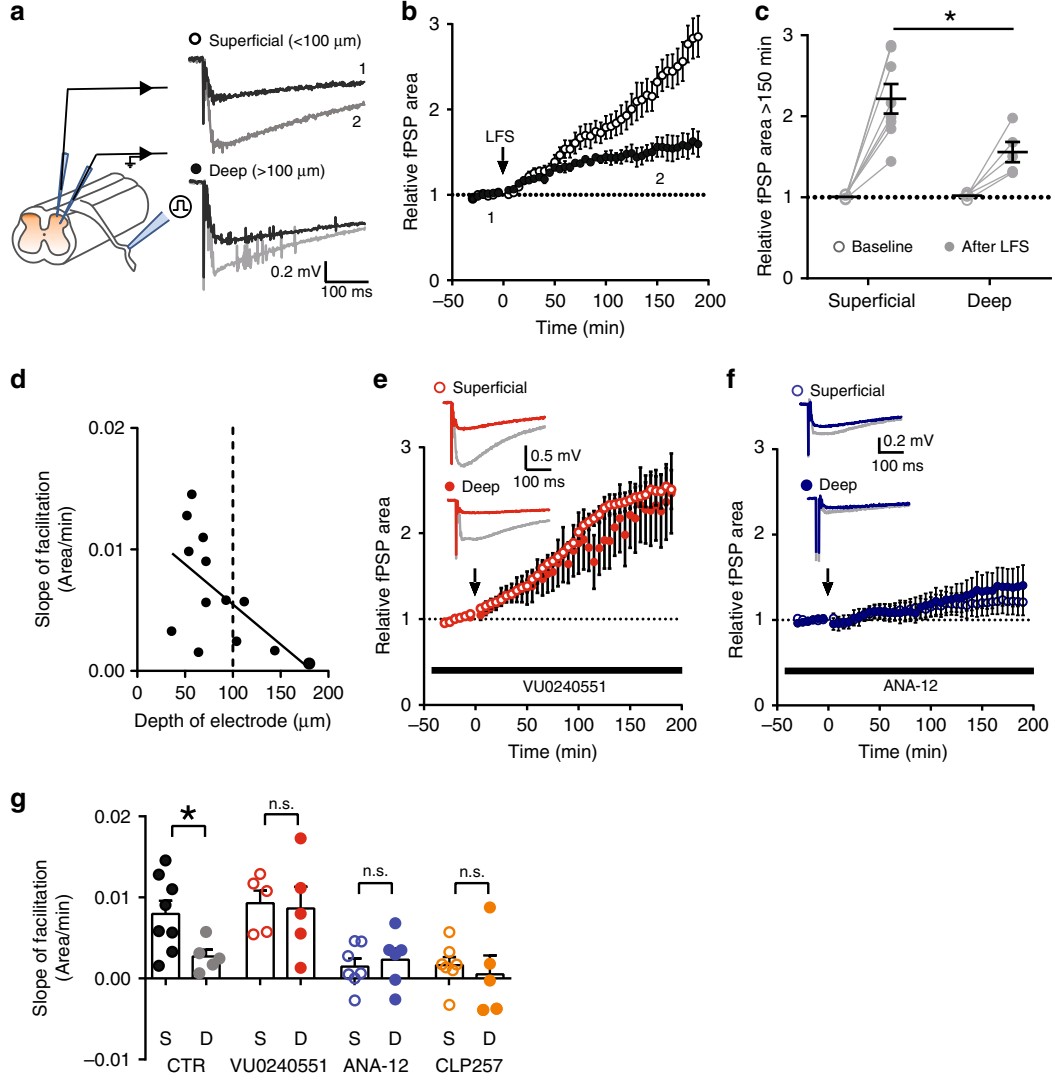

**Fig. 7 KCC2 levels shape synaptic plasticity across the superficial dorsal horn. a** Schematic representation of the experimental procedure (*left*). Field responses were evoked by dorsal root stimulation and recorded at different depths from the dorsal surface (*right*). Baseline responses (*black*) were facilitated after low-frequency stimulation (LFS; *gray*). **b** Time course of fPSP responses in superficial (*n* = 8) and deep recordings (*n* = 5). Arrow shows the time of LFS application. Number insets correspond to baseline and facilitated traces in **a**. **c** LFS produced a greater increase in fPSP responses in superficial than in deep recordings after 150 min (*t* = 2.6, *P* = 0.02). Individual experiments are depicted in gray. **d** Relationship between the rate of fPSP area increase (slope of LTF) and the distance of the recording electrode position from the external dorsal horn surface. The slope of LTF decreases as the recording electrode is placed deeper into the dorsal horn. **e** Blocking KCC2 activity with VU0240551 (10 μM) produced similar increase in fPSP responses after LFS in superficial (*n* = 5) and deep recordings (*n* = 5). Insets show representative traces of fPSP responses during baseline (*red*) and 150 min after LFS (*gray*). **f** LTF was stabilized in superficial recordings (*n* = 7), similar to deep (*n* = 6), when TrkB signaling was blocked with ANA-12 (1 μM). Insets show representative traces of fPSP responses during baseline (*blue*) and 150 min after LFS (*gray*). **g** Differences in slope of LTF between superficial (*black*) and deep (*gray*) recordings. VU0240551 (*red*) abolished the difference in slope of LTF between superficial and deeper recordings (unpaired *t*-test, *t* = 0.2, *P* = 0.8), while ANA-12 (*blue*) or CLP257 (5 μM, *orange*) treatments stabilized the slope of LTF in superficial recordings (ANA-12: *t* = 0.5, *P* = 0.6; CLP257: *t* = 0.5, *P* = 0.6). CTR control, fPSP field post-synaptic potential, LFS low frequency stimulation, LTF long-term facilitation. S superficial, D deep, n.s. not significant. Data are shown as mean ± S.E.M. *P < 0.05.

way-RM-ANOVA, $F_{interaction}$ = 0.6, *P* = 0.9; Supplementary Fig. 7e, f and Fig. 7g). These findings indicate that the propensity to display runaway facilitation in the most superficial part of the SDH is linked to low KCC2 activity resulting from on-going TrkB-dependent signaling.

**Uneven Cl⁻ transport impacts afferent-specific sensitization.** To address the impact of greater lability of inhibition and higher activity-dependent plasticity in LI, we took advantage of mice expressing channelrhodopsin-2 (ChR2) in two classes of afferents with differential projection patterns in the SDH. TRPV1 afferents

predominantly terminate in LI and outer LII, whereas MRGPRD afferents mainly terminate in LII in mice[64–66]. To selectively activate each class of afferents optogenetically, we crossed TRPV1-cre and MRGPRD-cre mice with floxed-ChR2-YFP. We confirmed that TRPV1 afferent fibers projected more superficially (overlapping with CGRP immunoreactivity), while MRGPRD fibers were mainly located deeper (overlapping the region of IB4 labeling; Fig. 8a–d). This provided a means to probe behavioral sensitization following activation of input to distinct SDH laminae. MRGPRD- and TRPV1- fibers were activated by sustained blue light stimulation (2 Hz, 5 min) to the plantar surface of the

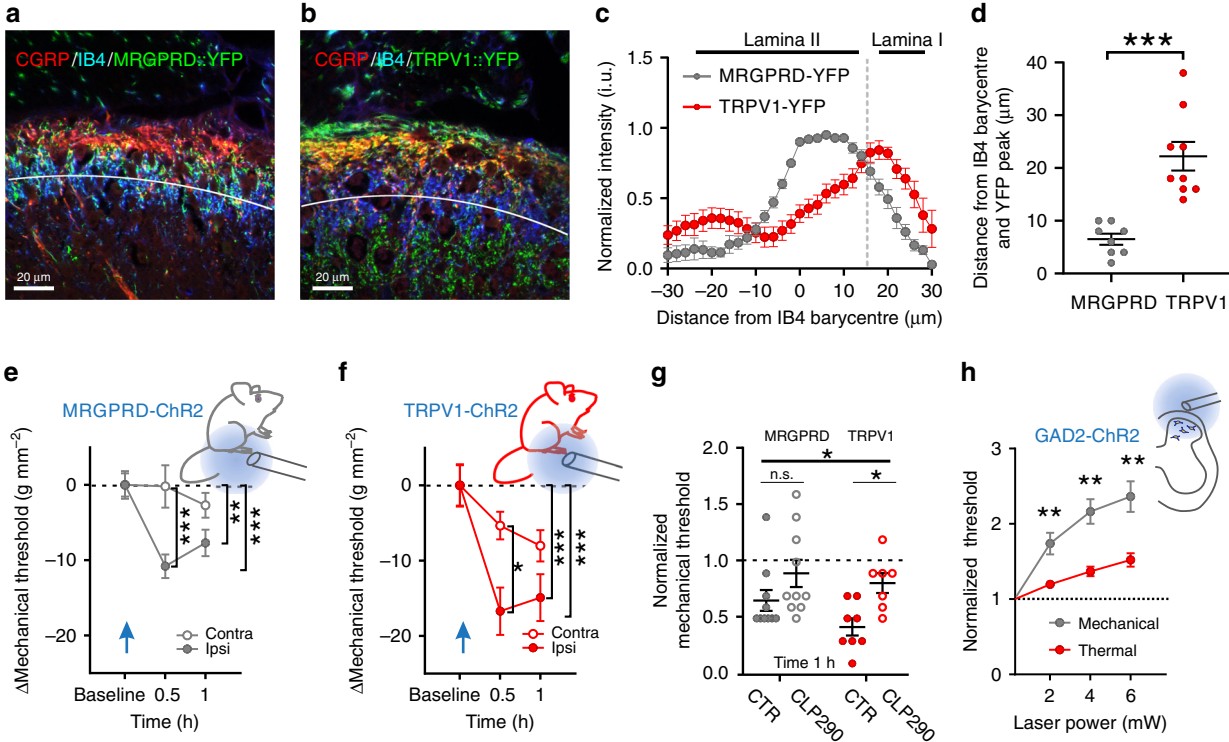

**Fig. 8 Cl⁻ homeostasis impacts mechanical and thermal nociceptive processing. a, b** Confocal images of the distribution of YFP + projections in the superficial dorsal horn of MRGPRD-ChR2 (**a**) and TRPV1-ChR2 (**b**) mice. Laminar boundaries were delineated by CGRP (*red*) and IB4 (*blue*) staining. The *white line* represents the IB4 barycentre in the SDH. **c** Distribution of YFP + fluorescence intensity from eight MRGPRD-ChR2 and nine TRPV1-ChR2 mice as a function of the IB4 barycentre (*white line* in **a, b**). Dashed line indicates the boundary between laminae I and II. **d** The peak of YFP-intensity associated with MRGPRD afferents (*gray*) is closer to the IB4 barycentre (LII) than TRPV1 afferents (*red, t = 5.1, P < 0.001*). **e, f** Time course of mechanical sensitization induced by blue light illumination (5 min, 2 Hz) of the hindpaw in MRGPRD-ChR2 (**e**, *n = 10*) and TRPV1-ChR2 (**f**, *n = 8*) mice. The insets schematically illustrate the experimental procedure. **g** Effect of orally administered KCC2 enhancer CLP290 (100 mg kg⁻¹) on laser-induced sensitization (*time 1 h = 1 h* after laser stimulation). CLP290 has no effect on MRGPRD-ChR2 mechanical sensitivity, while it prevents sensitization in TRPV1-ChR2 mice. **h** Effect of the activation of ChR2-expressing GAD2⁺ SDH interneurons by blue light stimulation on mechanical (*n = 13*, grey) and thermal (*n = 7*, red) sensitivity. Nociceptive threshold was measured at different laser powers and normalized to baseline. Inset shows a schematic diagram illustrating epidural stimulation of ChR2. i.u. intensity units, CGRP calcitonin gene-related peptide, IB4 isolectin B4, MRGPRD Mas-related G-protein coupled receptor member D, TRPV1 transient receptor potential vanilloid 1, ChR2 channelrhodopsin-2, YFP yellow fluorescent protein, Contra contralateral, ipsi ipsilateral, GAD2 glutamate decarboxylase 2. Data are expressed as mean ± S.E.M. *P < 0.05, **P < 0.01, ***P < 0.001.

left hindpaw[67]. Both MRGPRD- (n = 10) and TRPV1-ChR2 mice (n = 8) displayed mechanical hypersensitivity on the ipsilateral paw after stimulation (MRGPRD: Two-way-RM-ANOVA $F_{time} = 5.8$, $P = 0.006$, Fig. 8e; TRPV1: Two-way-RM-ANOVA $F_{time} = 11.9$, $P < 0.001$, Fig. 8f). However, sensitization in TRPV1-ChR2 was greater than in MRGPRD-ChR2 mice 1 h after paw stimulation (unpaired *t*-test, $t = 2.1$, $P = 0.049$). To exclude biases due the developmental shift in TRPV1/MRGPRD expression in transgenic mice[68], the experiment was repeated using viral delivery of Cre-dependent ChR2 in postnatal TRPV1-/MRGPRD-Cre mice (Supplementary Fig. 8). In these mice the pattern of expression of TRPV1-ChR2/MRGPRD-ChR2 afferents across the SDH was comparable, albeit better segregated, than that in the crossed transgenic mice (Supplementary Fig. 8a–d). In the viral-transduced mice, the time course of sensitization after stimulation of MRGPRD-ChR2 afferents lasted only 30 min (Supplementary Fig. 8e), while it lasted over 1 h in viral-transduced TRPV1-ChR2 afferents (Supplementary Fig. 8f). Collectively, these results show that sensitization induced through TRPV1-expressing primary afferent fibers is more intense and longer-lasting than that induced through MRGPRD fibers. Interestingly, enhancing KCC2 activity by administration of orally available CLP290[48], significantly attenuated sensitization from activation of TRPV1 afferents (two-way-ANOVA, $F_{treatment} = 9.7$, $P = 0.004$; Fig. 8g).

Thus, uneven strength of inhibition across laminae translates into differential sensitization to input from specific classes of nociceptive afferents.

**Modality-specific differential impact of inhibition.** As TRPV1 and MRGPRD afferents mainly encode thermal and mechanical inputs, respectively[64], we tested whether inhibitory strength differentially regulates these two sensory modalities. We used Gad2-Cre crossed with floxed-ChR2 mice to selectively activate spinal inhibitory interneurons by epidural optogenetics (Fig. 8h)[69]. This produced an intensity-dependent decrease of both mechanical (n = 13) and thermal (n = 7) sensitivity (increase in threshold; two-way-RM-ANOVA $F_{time} = 24.4$, $P < 0.001$). However, the relative analgesic effect was significantly smaller for thermal sensitivity (two-way-RM-ANOVA $F_{modality} = 10.0$, $P = 0.005$; Fig. 8h). Thus, weaker inhibition in LI may account for the inefficacy of inhibitory transmission in controlling nociceptive thermal input and constraining thermal sensitization.

## Discussion
Our data unveiled a TrkB-dependent gradient in KCC2 expression and function across the adult SDH. The ensuing differential ionic plasticity manifests itself in the form of more labile inhibition as well as a novel form of metaplasticity, expressed as

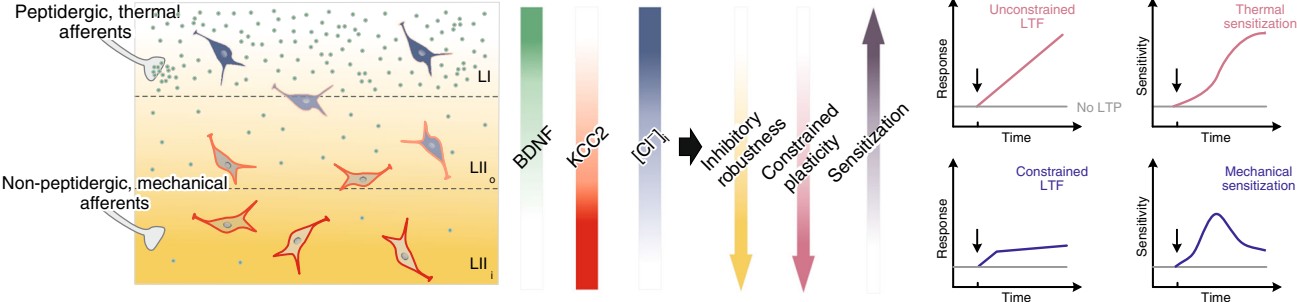

**Fig. 9 KCC2-dependent metaplasticity in the SDH.** The diagram schematically illustrates the impact of ionic plasticity in SDH on laminar differences in synaptic plasticity. Higher activation of TrkB in LI, as compared to LII, which likely follows a gradient of BDNF availability from afferent fibers (*green gradient and dots*), sets a gradient in KCC2 expression increasing from dorsal to ventral (*red gradient*), leading to higher activity-dependent Cl⁻ accumulation in LI (*blue gradient*) and more robust inhibition in LII (*yellow gradient*). Higher ionic plasticity in LI results in a runaway, unconstrained plasticity, while it is more and more constrained in deeper laminae (*purple gradient*). The lamina-specific differences in synaptic plasticity affects modality-dependent sensitization induced by a dominant LI input (TRPV1 afferents, encoding thermal stimuli) or a dominant LII input (MRGPRD afferents, encoding mechanical stimuli).

runaway synaptic facilitation. Such form of feedforward amplification may explain the propensity for nociceptive pathways to self-amplify input, so that the resulting aversive sensation can rapidly reach overwhelming proportions. This amplification process appears more dramatic for thermal input, which is primarily processed in LI.

The relationship between KCC2 activity and efficiency of inhibition is highly nonlinear[5]. Even small changes in Cl⁻-extrusion capacity deeply affect spatial and temporal summation of inhibitory inputs thus unsettling the control of firing activity[26,28]. Adopting adequate methods to measure fluctuations in $[Cl^-]_i$ is the necessary premise to unveil inconspicuous variations in KCC2 function[5]. Appropriate methods should consider the dynamic nature of the relationship between GABA$_A$R-/GlyR-mediated currents and KCC2[70]. The phenomenon known as ionic plasticity implies that the driving force of GABA$_A$ and glycine currents is continuously shaped by synaptic activity[5,26]. High frequency of inhibitory and excitatory input increases the Cl⁻ load in neurons and affects the amplitude of inhibitory currents according to the level of KCC2 activity[26].

We reported negligible differences in Cl⁻ homeostasis across SDH neurons when a low Cl⁻ load was imposed. Conversely, imposing a high Cl⁻ load through the recording pipette or via increasing synaptic activity was sufficient to unveil a gradient in KCC2 function across LI and II. This is strongly supported by past studies, suggesting that the impact of Cl⁻ transporters on Cl⁻ homeostasis is better investigated by imposing a Cl⁻ load[19,34,71]. This should be kept in mind when interpreting results from previous studies. Recent evidence has argued in favor of a predominant action of local impermeant anions through the Donnan effect in setting $[Cl^-]_i$[24]. These experiments were however performed in the presence of limited synaptic activity, suggesting that the relative contribution of KCC2 was likely underestimated. Thus, while impermeant anions appear to set $[Cl^-]_i$ and account for its intrinsic variability among neurons[24], our results reveal that KCC2 activity is critical to maintain the Cl⁻ gradient under Cl⁻ load conditions. Cl⁻-extrusion capacity therefore determines how post-synaptic inhibition is shaped by network activity.

In spinal nociceptive pathways, BDNF is released by peptidergic primary afferent fibers in the SDH[72]. BDNF-TrkB signaling represents a common signaling pathway shaping plasticity in different regions of central nervous system at both excitatory and inhibitory synapses[20,73]. Activation of TrkB signaling also represents a canonical mechanism regulating inhibition in SDH neurons[20,46], although the direction of this effect highly depends on the specific intracellular pathways involved[21]. While TrkB signaling enhances KCC2 activity early in development and during certain phases of post-traumatic injuries[74], our results indicate that on-going TrkB signaling negatively regulates KCC2 in the mature SDH.

Interestingly, downregulation of KCC2 has been shown to promote TrkB-dependent potentiation of NMDA receptors[47,75]. Here, we identify a novel form of metaplasticity directly associated with the uneven level of KCC2 across SDH: low KCC2 in LI is causally linked to runaway LTF, which was absent in LII. Weak KCC2 appeared both necessary and sufficient to explain this on-going facilitation. This phenomenon likely reflects a persistently enhanced excitatory drive which continuously challenges inhibition[76]. Our data also indicate that synaptic plasticity is mostly shaped by post-synaptic differences in Cl⁻ homeostasis rather than by differences in presynaptic arrangements[77]. While modulation of KCC2-dependent plasticity may result from descending inputs[30,78], we found that the trans-laminar KCC2 gradient is both present in vivo and maintained after severing descending input in our ex vivo preparations, consistent with the postulate of on-going TrkB signaling engaged by activity in peptidergic afferents[79]. Thus, opposing gradients in TrkB signaling and KCC2 activity come together to shape the strength of inhibition and LTF behavior in SDH (Fig. 9). In turn, activity-dependent plasticity of inhibition promotes metaplasticity at excitatory synapses[30,31,59,80].

High levels of KCC2 lead to hard-wired neuronal circuits that are finely tuned by synaptic inhibition. Conversely, low KCC2 is associated with enhanced cross-talk between excitatory synapses[59] and a higher propensity to undergo plasticity (metaplasticity)[30].

The gradient in KCC2 activity matches the functional organization of SDH in which thermal and mechanical stimuli are processed in distinct sublaminae: thermal input in LI and outer LII, mechanical input in the inner LII and deep dorsal horn[64,81]. Thermal and mechanical pain is thus associated with defined subregions, displaying different levels of Cl⁻-mediated control. Interestingly, intrathecally delivered GABA/glycine antagonists have dramatic effects on tactile sensitivity, with little consequences on thermal behavior[82,83]. Similarly, we found that optogenetic silencing of inhibitory interneurons in the SDH causes mechanical sensitization, with minimal effects on thermal sensitivity[69]. Optogenetic activation of SDH inhibitory interneurons induces greater mechanical than thermal analgesia, and selective activation of TRPV1 afferents produces greater sensitization, compared to MRGPRD afferents. Recent work highlighted that impaired Cl⁻ regulation has a greater impact on excitatory

vs. inhibitory neurons because excitability in the former rely more heavily on inhibition[84]. Altogether, these observations support the prediction that the functional impact of disinhibition is larger where inhibition is more robust.

In pathological conditions that involve KCC2 hypofunction[13,85], disinhibition in LII will change local hard-wired circuits into plastic circuits, thus leading to maladaptive plasticity within mechanical pathways. If the same applies to humans, it may explain why mechanical allodynia represents one of the more prominent clinical symptoms of traumatic painful neuropathy[86].

In conclusion, our results pinpoint the regulation of Cl⁻ homeostasis as a critical mechanism that defines the behavior of synaptic plasticity in the SDH, thus shaping adaptive mechanisms in physiological settings and maladaptive drift in pathology.

## Methods

**Animals**. All experimental procedures have been performed in accordance with guidelines from the Canadian Council on Animal Care and approved by the committee for animal protection of Université Laval (CPAUL; authorization number: 2018-027-1 and 2018-026-1).

Functional and morphological experiments were performed on adult male Sprague Dawley rats (> 60 days; Charles River Laboratories) or adult male C57Bl/6 mice (> 60 days; Charles River Laboratories). Optogenetics studies were performed on (i) adult (3-months-old, male) Ai39 (RCL-eNPHR3.0-EYFP) mice (The Jackson Laboratory, 014539); (ii) adult (3–7 months) MRGPRD–/TRPV1-ChR2 mouse lines generated by crossing Cre mice (MMRRC, stock No. 036118-UNC) or TRPV1-Cre mice (The Jackson Laboratory, 017769) and Ai32 (ROSA-CAG-LSL-ChR2) mice (The Jackson Laboratory, 012569); (iii) adult (> 3 months) GAD2-ChR2 mouse line generated by crossing GAD2-IRES-Cre line (The Jackson Laboratory, 010802) with Ai32 (ROSA-CAG-LSL-ChR2) mice line (The Jackson Laboratory, 012569).

**Virus injection**. Mice expressing eNPHR3.0 in SDH neurons were generated by intraspinal injection of 200 nl of AAV8-CMV-Cre virus (AAV8.CMV.HI.eGFP-Cre. WPRE.SV40, UPenn Vector Core, AV-8 PV2004) into adult Ai39 mice. Briefly, mice were anesthetized with 2.5 to 3% isofluorane and L3 to L5 spinal cord was exposed by carefully removing T13 vertebrae spinous process without laminectomy. Virus were pressure ejected into the spinal parenchyma at a depth of 100 μm via a glass pipette connected to a nanoinjector (Micro 4, WPI) at a rate of 1 nl s⁻¹. The injection was carried out four times bilaterally (two times per side) on each mouse. Mice were allowed to recover from intraspinal injection for 4 weeks prior to experimentation. After the experiment, the spinal cords were fixed in paraformaldehyde 4% overnight for confirmation of virus expression.

For conditional channelrhodopsin-2 (ChR2) expression in sensory neurons, either MRGPRD-Cre or TRPV1-Cre pups (P5) were intraperitoneally injected with 20 μL of a Cre-dependent AAV9 (FLEX-rev-ChR2(H134R)-mCherry, $1.8 \times 10^{13}$ vg mL⁻¹, Addgene #18916, lot v22123). This method has been shown to transfect DRG neurons only[69].

**In vivo pharmacological blockade of TrkB receptors**. The selective TrkB antagonist N-[2-[[(Hexahydro-2-oxo-1H-azepin-3-yl)amino]carbonyl]phenyl]benzo [b]thiophene-2-carboxamide (ANA-12; Sigma, catalog#SML0209) was dissolved at the final concentration in sterile saline solution with 1% of dimethyl sulfoxide. Twelve rats received either a single i.p. injection of ANA-12 (0.5 mg kg⁻¹)[51] or vehicle. Animals were sacrificed after 4 h for immunohistochemical analysis.

**Preparation of spinal cord slices**. Spinal cord slices were prepared as previously described[40]. Rats were anaesthetized with ketamine/xylazine (i.p., 8.75/1.25 mg per 100 g). Rats were briefly transcardially perfused with ice-cold oxygenated (95% O₂, 5% CO₂) sucrose-based artificial CSF (S-ACSF) solution containing the following (in mM): 252 sucrose, 2.5 KCl, 2 MgCl₂, 2 CaCl₂, 1.25 NaH₂PO₄, 26 NaHCO₃, 10 glucose, and 5 kynurenate. After decapitation, spinal cords were removed by hydraulic extrusion and immerged in the same ice-cold solution. Lumbar enlargements were isolated and 300 μm-thick parasagittal (for electrophysiology) or transverse (for Cl⁻ imaging) slices were obtained with a Leica vibratome. Slices were allowed to recover for 30 min at 34 °C in an immersion chamber in oxygenated ACSF.

**Electrophysiology**. After recovery in oxygenated ACSF, slices were transferred to a recording chamber and continuously superfused with oxygenated ACSF (2–3 mL min⁻¹) containing the following (in mM): 126 NaCl, 2.5 KCl, 2 MgCl₂, 2 CaCl₂, 1.25 NaH₂PO₄, 26 NaHCO₃, 10 glucose. Neurons were visually identified

under a Zeiss Axioscope equipped with infrared differential interference contrast (IR-DIC) and 40x water-immersion-objectives.

LI was identified by visual inspection as a narrow dark band of gray matter with a typical reticulated appearance less than 50 μm away from the dorsal white matter, LII as a wider translucent band below LI[40]. Since laminar width considerably varies according to the distance from the midline, correct laminar localization was confirmed by immunostaining (see below).

Patch pipettes (5–7 MΩ) were pulled on a horizontal puller (P-97; Sutter) and filled with the following intracellular solution (in mM): 115 K-methylsulfate, 25 KCl, 2 MgCl₂, 5 KCl, 10 HEPES, 4 ATPNa, 0.4 GTPNa, 0.1% Lucifer-Yellow (LY, Sigma), pH 7.2 adjusted with KOH. For low pipette [Cl⁻] recordings, KCl was replaced with K-methylsulfate to obtain 9 mM [Cl⁻]. For perforated-patch experiments, gramicidin was freshly prepared as a stock solution of 60 mg mL⁻¹ in dimethyl sulfoxide and dissolved immediately before the experiment in the intracellular solution at the final concentration (30 μg mL⁻¹) by brief sonication. In experiments designed to quantify the effect of capsaicin on excitatory and inhibitory post-synaptic currents (EPSCs and IPSCs) recordings, KCl and K-methylsulfate in the intracellular solution were replaced with Cs-methanesulfonate (140 mM). EPSCs were isolated at –60 mV and IPSCs at 0 mV.

Whole-cell and excised patch recordings were obtained using an Axopatch-200B amplifier (Molecular Devices), perforated-patch recordings were obtained using a MultiClamp 700B amplifier (Molecular Devices). Data were filtered at 4–5 kHz, digitized and acquired using the Strathclyde electrophysiology software WinWCP (courtesy of Dr. J. Dempster, University of Strathclyde, Glasgow, UK) or pClamp 10.2 software (Molecular Devices).

Passive and active membrane properties were recorded. The resting membrane potential was measured immediately after establishing the whole-cell configuration. Only neurons with a resting potential more negative than –50 mV and stable access resistance during the recording were included for subsequent analysis. Membrane potentials in whole-cell recordings were corrected off-line for liquid junction potential (8 mV for high [Cl⁻] pipettes and 9 mV for low [Cl⁻] pipettes). Electrophysiological data were analyzed with Clampfit 10.2 (Molecular Devices).

**$E_{GABA}$ measurement**. GABA (1 mM) was dissolved into a HEPES-buffered ACSF and applied by brief puffs (30 ms) from a patch pipette with a minimum interval of 20 s between applications. GABA responses were obtained in voltage clamp at increasing holding potentials (in 12.5 mV steps) in the presence of tetrodotoxin (TTX; Alomone lab, 1 μM), D(−)-2-amino-5-phosphonovaleric acid (APV; 40 μM, Sigma) and 6-cyano-7-nitroquinoxaline-2,3-dione (CNQX; 10 μM, Sigma) to block unwanted conductances. GABA I–V curves were obtained by averaging three responses for each voltage step and $E_{GABA}$ was extrapolated from the derived linear equation as the x-axis intercept. In absence of pharmacological treatments, $E_{GABA}$ in individual neurons was stable over time (–48 mV ± 4.7 at the beginning of the experiment vs. –46 mV ± 3.7 after 10–15 min, $n = 3$). [Cl⁻]ᵢ was calculated from $E_{GABA}$ by inverting the Goldman-Hodgkin-Katz-(GHK) equation assuming a permeability ratio between Cl⁻ and HCO₃⁻ anions of 0.25, with a [HCO₃⁻] of 26 and 16 mM in the extracellular and intracellular solutions, respectively[87] and a room temperature of 25 °C.

**Repetitive stimulation of inhibitory transmission**. Recordings were performed in whole-cell configuration with low [Cl⁻] pipette containing EGTA (0.5 mM) in the presence of CNQX (10 μM) and APV (40 μM) to block AMPA/Kainate and NMDA receptor-mediated currents. Evoked IPSCs (eIPSCs) were elicited by electrical stimulation (100 μA, 200 μs) delivered focally via a patch micropipette placed in the vicinity of the recorded cell, as described[40]. Trains of stimuli (25 pulses—20 Hz) were delivered every 20 s at 0 or –90 mV. The average of ten consecutive stimulations was used for subsequent analysis. Peak amplitude after each stimulation was normalized to the first eIPSC. Normalized amplitudes were binned by averaging three consecutive points every five stimuli. Data were fit with a mono-exponential decay curve.

Drug treatments to block TrkB receptors by ANA-12 (1 μM)[51] or enhance KCC2 activity by CLP257 (5 μM)[10,15,49] were performed by pre-incubating slices with the drugs for at least 2 h prior to recording. At this concentration, CLP257 selectively enhanced KCC2 without unspecific effects on GABA_ARs. All the drugs were continuously bath applied during the recording sessions.

**Laminar localization of recorded neurons**. After recording, spinal cord slices containing LY-injected neurons were fixed for 30 min with 4% paraformaldehyde in phosphate buffer (PB; 0.1 M, pH 7.4). After several washings in phosphate buffered saline (PBS; 0.05 M, pH 7.4), slices were pre-incubated in PBS with 1% normal goat serum for 30 min and then incubated overnight at 4 °C in rabbit anti-NK1 1:10000 (Sigma) for LI neurons, rabbit anti-PKCγ (1:100, Santa Cruz Biotechnology) or biotinylated isolectin B4 (1:1000, Sigma) for LII neurons. Slices were then washed in PBS and incubated for 3 h with anti-rabbit AlexaFluor 495 1:500 (Molecular Probes, Carlsbad, CA) or 1 h with Extravidin-Cy3 1:1000 (Sigma). Slices were finally mounted in Vectashield H-1000 mounting medium (Vector Lab). Confocal laser scanning microscopy was performed using a Zeiss LSM 510 confocal microscope with a 63x oil lens.

**Fluorescence lifetime imaging microscopy (FLIM)**. Transverse rat spinal cord slices were labeled in continuously oxygenated (95% O₂, 5% CO₂) ACSF containing 5 mM of the Cl⁻ indicator MQAE (N-6-methoxyquinolinium acetoethylester, Molecular Probes) for 30 min at 34 °C. Slices were then transferred to a perfusion chamber (2 mL min⁻¹) and perfused with oxygenated ACSF at 34 °C for 10 min to allow extracellular MQAE washout. MQAE fluorescence was then measured as described previously[19]. Briefly, the sample was excited using a femtosecond-pulsed Ti-Sapphire laser (80 MHz repetition rate) tuned at 760 nm, imaged using a 40x water-immersion objective (Zeiss, 0.8 N.A.), and fluorescence acquired through both a band-pass filter (469/35 nm, Semrock) and a short-pass filter (750 nm, Semrock), onto a PMC-100-1 photosensor (Becker & Hickl GmbH, Germany) for photon counting.

Fluorescence lifetime measurements were calculated using custom MATLAB software (The MathWorks Inc). Briefly, photon histograms were obtained for each neuron identified as individual ROIs. Mono-exponential decays ($y = y_0.e^{-t/\tau}$, with $\tau$ the fluorescence lifetime), convolved with the measured instrument response function were then fit to these traces. The instrument response function of the detection path was acquired using an 80 nm gold nano-particle suspension (Sigma-Aldrich) to generate second-harmonic signal. A minimum of ten FLIM images (10 s of acquisition for each) were acquired.

The dorsal white matter border was defined as the outer limit of the MQAE labeling of LI. Lifetime values of SDH neurons were measured under two conditions: (i) TTX (1 μM), bicuculline (20 μM, sigma), strychnine (1 μM, sigma), CNQX (10 μM, sigma), APV (40 μM, sigma) to block synaptic activity ($N = 14$ slices from two rats) and (ii) capsaicin (2 μM, Sigma) to enhance synaptic activity ($N = 14$ slices, from three rats). MQAE fluorescence lifetime values were converted to [Cl⁻] in mM according to the Stern–Volmer equation ($t_0/t = 1 + Ksv$ [Cl⁻]), where $t_0$ is the fluorescence lifetime in 0 mM Cl⁻ and Ksv is a coefficient of Cl⁻ sensitivity of MQAE, which is equal to 32 M⁻¹, as reported[26]. Estimated [Cl⁻] was plotted against the distance of the cell soma to the edge of white matter.

**Post-synaptic field potential (fPSP) recordings**. Mice were anesthetized with urethane (i.p., 2 g kg⁻¹) and perfused intracardially with ice-cold oxygenated (95% O₂ and 5% CO₂) S-ACSF containing (in mM): 50 sucrose, 92 NaCl, 5 KCl, 0.5 CaCl₂, 7 MgCl₂, 15 glucose, 26 NaCO₃, 1.25 NaH₂PO₄, 1 kynurenate. The lumbar spinal column was rapidly obtained and a laminectomy performed in cold S-ACSF. The ventral roots and connective tissue were removed from the spinal cord and a lumbar segment with attached dorsal roots was placed in oxygenated ACSF at room temperature for 1 h before recording.

Synaptic potentiation of fPSPs in the superficial dorsal horn was produced as previously described[60]. Briefly, fPSP responses were recorded with borosilicate glass electrodes filled with ACSF (3–5 MΩ) inserted into the dorsal root entry zone of the spinal cord. A suction electrode was placed in a dorsal root to deliver electrical stimulation. Test pulses were presented every 60 s. Signals were amplified with a Multiclamp 700B amplifier (Molecular Devices), digitized with a Digidata 1322 A (Molecular Devices), and recorded using pClamp 10 software (Molecular Devices). Data were filtered at 1.6 kHz and sampled at 10 kHz. Stimulus intensity was determined for each experiment and adjusted to evoke 50% of the maximum fPSP. After a stable baseline recording (30 min), potentiation of fPSP responses (long-term facilitation, LTF) was induced by low frequency stimulation of the dorsal root (LFS; 2 Hz, 2 min). In drug perfusion tests, the whole spinal cord tissue was treated for at least 1 h before recording and during the whole recording session. In Cl⁻ loading with eNpHR3.0, after a stable baseline recording of fPSPs was obtained, the spinal cord was continuously illuminated with a Ar-Kr laser at 568 nm (4–5 mW). After 10 min of continuous light, LFS stimulation was administered to dorsal roots to induce potentiation; the light was kept on throughout the rest of the recording period to maintain a continuous Cl⁻ load.

Data were analyzed using Clampfit software (Molecular Devices). The area of fPSPs relative to baseline was measured from 0 to 800 ms after the onset of the fPSP. Electrode depths from the dorsal surface of the spinal cord were measured with an MPC-200 micromanipulator (Sutter Instrument Company).

Recordings were considered superficial when the recording electrode was placed at no more than 100 μm from the dorsal spinal cord surface and deep when the electrode was placed at more than 100 μm. Superficial and deep recordings were grouped together.

**Immunohistochemistry**. Rats and mice were anaesthetized with equithesin i.p. (3 mg per 100 g body weight) or ketamine/xylazine i.p. (8.75/1.25 mg per 100 g). Animals were perfused transcardially with 4% paraformaldehyde in 0.1 M PB (pH 7.4) for 30 min. Spinal cord segments L4–L5 were collected and postfixed for 60 min in the same fixative and cryoprotected in 30% sucrose in 0.1 M PB overnight at 4 ºC. Transverse sections were cut at 25 μm on a sledge freezing microtome Leica SM2000R (Leica Microsystems). Sections were permeabilize in PBS (pH 7.4) with 0.2% Triton (PBS + T) for 10 min, washed twice in PBS and incubated for 24 h at 4 °C in primary antibody mixtures (see below) diluted in PBS + T containing 4% normal donkey serum. After washing in PBS, the tissue was incubated for 2 h at room temperature in a solution containing a mixture of appropriate fluorochrome-conjugated secondary antibodies diluted in PBS + T (pH 7.4) containing 4% normal donkey serum. Lastly, sections were mounted on gelatin-subbed slides (Fisherbrand), allowed to dry overnight at 4 °C and cover-slipped using Aquapolymount (Polysciences).

A polyclonal IgG anti-KCC2 antibody raised in rabbit against a His-tag fusion protein corresponding to residues 932–1043 of the rat KCC2 intracellular C-terminal was used for this study (1:1000; Millipore-Upstate, catalog #07-432)[88,89]. This immunogen is highly specific for KCC2 and does not show any sequence homology with other KCCs or CCCs. CGRP (calcitonin gene-related peptide; mouse anti-CGRP 1:2000; Sigma catalog #C 7113) and IB4 (AlexaFluor 488-conjugated IB4, 1:200, Invitrogen, catalog #I21411, Carlsbad, CA) staining were used to identify laminae I and II in the L4–L5 lumbar segments of adult rats. Antibodies and IB4 have been successively omitted to test for any possible cross-reactions.

**Confocal microscopy and image acquisition**. Images were obtained with an Olympus FV1000 (Olympus America Inc.) confocal laser scanning microscope (CLSM) with a 60x plan-apochromatic Apo oil immersion objective (NA 1.4) using dichroic filter FV-FCBGR 488/543/633. Each fluorophore was imaged sequentially to minimize channel bleedthrough and different emission filters (Chroma) were used for different fluorophores (510IF for Alexa488, 605BP for Cy3 and 660IR for Alexa647). An optimal setting of the laser power and PMT (PhotoMultiplier Tube) voltage was chosen to minimize pixel saturation, photobleaching and to make sure that the collected intensities were in the linear range of the PMT. The CLSM settings were kept constant for all comparable samples and controls (Laser power, filters, dichroic mirrors, polarization voltage, scan speed) so that valid comparisons could be made between KCC2 intensity measurements from different images (12-bits, 2048 × 2048 pixels pictures with pixel size of 0.103 μm). For all pixel intensities measured, a constant value, defined as the inherent noise of the photomultiplier tube (PMT) and computed as the mean intensity value of a region where no sample is present, was subtracted.

**Image segmentation and analysis**. Fluorescence confocal images of different markers were acquired to delineate the functional laminae in the spinal cord. As described earlier[42,63], IB4 and CGRP fibers are used to distinguish the different laminae. Algorithms were developed to quantitatively and adequately compare different spinal cord slices obtained from different animals using self-made algorithms in MATLAB. Four distinct but complementary analytical approaches were used to quantify the KCC2 distribution in this study. Fluorescence intensity is linearly proportional to emitter concentration over a large dynamic range. This fact allowed many quantitation of membrane-protein-distributions within well identified regions over wide fields of view[52,53,90–92]. It is important to note that incomplete labeling will indeed affect the total fluorescence intensity measured, but the fluorescence signal will still be proportional to the number of receptors targeted by the fluorescent probes[93–95]. Prior to the analyses presented below, the intensity of the immunostaining background noise was defined. For each image, the average intensity of a region where KCC2 is known to be absent (i.e., white matter of the spinal dorsal horn) was measured and subtracted to the whole image to exclusively calculated in the KCC2(+) subregion. In all the analytical approaches described below, the quantification of KCC2 immunofluorescence in the SDH was limited to the first 80 μm from the dorsal white matter. Within this region, LI is confined to the first 20–25 μm from the dorsal border.

For trans-laminar profile intensity analysis, the region of interest (ROI) was manually outlined along laminae I and II[42]. The center of the IB4 region was then calculated by a barycentric intensity weighted analysis of order 4 for each column of the image. To obtain a continuous layer, a smoothed center quadratic curve is then fit from the IB4 barycentric calculations. This quadratic curve (Fig. 4b) defines the origin of the IB4 axis projection. The minimum distance of every point in the image was calculated and the pixels being on the most superficial part of the spinal cord slice are arbitrarily defined as positive values on the IB4 axis projection whereas, the deeper pixels were defined as negative values on the IB4 axis projection. For each image, the average KCC2, IB4, and CGRP intensity were plotted as a function of the distance of the IB4 barycentric origin (Fig. 4b, c).

The membrane analysis by global index (MAGI) considers that the SDH corresponds to a complex dense network of cells and fibers. We previously developed an algorithm[48,49] to isolate the KCC2 membrane immunostaining from the KCC2 intracellular in the SDH. Indeed, even if the cell membrane is precisely delineated, the measurement would still be heavily tainted by the presence of intracellular KCC2 due to the optical resolution as determined by the point spread function. Using the MAGI approach, we defined an index that reflects the global membrane KCC2 intensity and hence, is not based on manual selection. The intensity of the intracellular KCC2 immunostaining was defined in regions identified as cytoplasmic portions of KCC2-positive neurons. Finally, LI and LII of the SDH were then delineated and the average KCC2 pixel intensity was calculated. To obtain the KCC2 intensity corresponding to the membrane staining, the average intracellular KCC2 intensity value was subtracted to this global average KCC2 intensity in the chosen region. Membrane KCC2 intensity index was measured for every rat and the values were averaged. This index is robust and global because it includes many neuronal cell bodies and dendrites and does not depend on arbitrarily visually selected neurons (Figs. 4f and 5d).

Individual trans-membrane intensity profiles were also manually delineated in randomly selected SDH neurons for trans-laminar user-defined membrane

intensity analysis. KCC2 neuronal membrane intensities were reported along the dorso-ventral axis of the SDH, setting white matter origin as a zero. This method allowed countering the potential bias of artificial discrepancies in the intensity of KCC2 staining due to lower neuronal densities and cell body diameters that could differ in different SDH laminae (Fig. 4d).

The membrane analysis of sub-cellular profile intensity (MASC-π) method is based on an already published algorithm used to detect receptor membrane internalization[47]. This technique was developed to reduce bias that can arise from user interventions. For each confocal image (randomly and blindly selected), the membrane of neuronal cells was manually delineated. For each pixel in the region of interest, the distance to the closest membrane segment was calculated. Using this distance map, the mean pixel intensity and standard deviation of KCC2 fluorescence signal were quantified as a function of the distance to the neuronal membrane (defined as zero). Positive values correspond to neuronal intracellular space (Figs. 4e and 5c).

**Pre-embedding electron microscopy with FluoroNanogold**. Free-floating spinal cord vibratome sections were pre-incubated in PBS–5% NGS for 30 min at room temperature and then incubated overnight at 4 °C with chicken anti-full length-TrkB primary antibody (1:500; Promega Corporation, Cat# G1561). After washings in PBS, sections were incubated for 1 h with anti-chicken IgY biotinylated secondary antibody (1:250; Vector), and for an additional hour with AlexaFluor 488-Fluoronanogold™-Streptavidin (1:100; Nanoprobes). After observation of immunostaining at the fluorescence microscope, TrkB-labeled sections were postfixed 1 h in osmium ferrocyanide (1 volume of 2% aqueous osmium tetroxide: 1 volume of 3% potassium ferrocyanide), stained 1 h with 1% uranyl acetate in maleate buffer, dehydrated in increasing concentrations of ethanol, and embedded in Araldite. Ultrathin sections were cut with an ultramicrotome and collected on uncoated nickel grids (200 mesh). Grids were rinsed in distilled water and gold particles were intensified by a Gold Enhancement Kit (Nanoprobes) for 15 min to increase the size of the gold tag. The sections were finally counterstained with uranyl acetate and lead citrate before observation with a Philips CM10 electron microscope.

**Electron microscopy analysis**. Counts of TrkB immunoreactive cell bodies were performed on randomly selected ultrathin sections obtained from adult rats ($n = 3$) within individual 90 x 90 μm squares of 200 mesh grids by choosing those fields where the pial surface was in contact with one of the grid bars and moving perpendicularly toward the depth of the SDH. LI has been distinguished from LII by the presence of abundant small myelinated fibers, almost entirely absent in LII[96]. Only TrkB receptors clearly localized on cell bodies in LI ($n = 32$) and LII ($n = 33$) were considered for this analysis. TrkB receptors localized on the axon terminal were excluded as they are mainly associated with primary afferents[50]. Neuronal dendrites were also excluded as the laminar localization of their respective cell bodies was uncertain.

Using a custom-made MATLAB algorithm, the intensity weighted centroid positions of TrkB gold-intensified particles were manually selected from electron microscopy micrographs. The precise positions of all single TrkB particles from a given cell body were calculated. A histogram of all possible distances between any two particles being part of the same cell body was built. An all-distance histogram was obtained for both LI (831 detections) and LII (745 detections). Using the information obtained on the distances between particles, and assuming each particle being associated with a single receptor, detected TrkB receptors were split into oligomeric and monomeric groups adopting 65 nm as cutoff distance. The cutoff value was chosen as it falls below the maximum distance at which a pair of primary antibody/secondary antibody/intensified gold particle complexes targeting dimeric receptors can be found (~80 nm). Thus, TrkB receptor pairs that were measured to be less than 65 nm apart were considered to be part of a same receptor cluster, likely representing an oligomeric group. By transitivity, if the distance of receptors A & B is smaller than 65 nm and the distance of receptors B & C is smaller than 65 nm, then receptors A & C are considered to be part of the same oligomeric group even if the distance between them is larger than 65 nm. Other isolated particles were defined as "monomers", although we cannot exclude that some of them may be undetected oligomers. However, if such an error occurs, it would similarly affect both laminae. The number of receptors in each group was then established when all distances smaller than 65 nm were considered.

**Optical stimulation of primary afferent fibers**. Adult MRGPRD-ChR2 or TRPV1-ChR2 mice (either obtained by crossing transgenic animals or by viral injection; see above) were anesthetized with isofluorane 2% and an optical fiber (200 μm core, numerical aperture of 0.39) was placed at ~0.5 cm from the skin of the left hindpaw. Primary afferents were stimulated at low frequency (2 Hz, 10 ms pulses, 18 mW) for 5 min. CLP290 (100 mg kg⁻¹) was dissolved in 20% HPCD and given orally 2 h before sensitization[48]. At this dose, CLP290 was previously shown to increase KCC2 expression in the CNS[48,97,98].

At the end of the experiment, animals were perfused (as above described) and the distribution of targeted afferents analyzed. The YFP or mCherry signal in MRGPRD- and TRPV1-ChR2 mice was analyzed using Trans-laminar profile intensity analysis (see Image segmentation and analysis, above). Images were

obtained with a LSM880 (Zeiss) with a 63x oil objective (8 bits, 2048 × 2048 pixels, with a 0.066 pixel μm⁻¹ resolution).

**Optical stimulation of dorsal horn inhibitory interneurons**. GAD2-ChR2 mice were anesthetized with isoflurane 2%. Epidural fiber implantation was done as described previously[69]. Briefly, neck muscles were cut along the midline and retracted to provide access to the C1 vertebra. The atlantooccipital membrane was pierced and the optical fiber was pulled through this incision while holding the head of the animal at a 45° downward angle. The fiber placement was confirmed during dissection after experiments. Mice were allowed to recover from surgery at least 1 week before experimentation.

The day of the experiment, mice were acclimated in the testing apparatus for at least 1 h before testing. A laser source (Laser Diode Fiber Light Source, Doric Lenses) was used to deliver 450 nm light stimulations (20 Hz, 10 ms pulses) through an optical fiber cable (MFP_100/125/900-0.37_2m_FC-MF-2.5, Doric Lenses). The current was adjusted on the laser source between each implant to get a constant light power at the end of the fibers. The implant consisted of a 4.2 cm long optical fiber with a diffusive tip (MMF_POF_240/250-0.63_8 cm_DFL, Doric Lenses) and a stainless ferrule (SF270-10, Thorlabs) embedded in a dental cement base. The laser was turned on at least 5 s before each test round and turned off between rounds. Each round was separated by at least 5 min. Light intensities (0, 2, 4, 6 mW) were applied in ascending order.

**Behavioral tests**. Animals were placed on a von Frey apparatus and allowed to recover from anesthesia. Mechanical threshold was tested using a modification of the simplified up-down method[99]. A test round started with filament #5 (0.16 g; for sensitization experiments) or #7 (0.60 g; for mechanical threshold in experiments on GAD2-ChR2 mice) and progressed to higher or lower filament value depending on the animal's response. Each animal went through two test rounds for each paw at each experimental condition. Mechanical threshold is expressed as pressure (g mm⁻²) by taking into consideration the filament cross-sectional area.

Thermal threshold was tested using Hargreaves method. Briefly, animals were put on a warm glass panel (Model 400, IITC) and a radiant light was pointed at their paw. Thermal threshold is defined as the time to withdraw the paw after light onset. The cutoff point was set at 25 s to avoid injury. The animals were tested two times on each paw at each light intensity.

**Statistical analysis**. Statistical analysis was performed with GraphPad Prism 8 (GraphPad Software). Data were tested for normality with a Kolmogorov–Smirnov test. Paired or unpaired $t$-tests were used for comparing matched or unmatched groups, respectively. Differences between paired (repeated measures-RM) and unpaired values in multiple comparisons were compared with one-way or two-way analysis of variance (ANOVA) followed by Bonferroni post-hoc tests. $F$-test was used to analyze mono-exponential decay fittings to compare plateau and $K$ constant and linear fittings to compare slopes. The slope of LTF was obtained by linear regression of individual experiments and was assessed as a function of electrode depth by Pearson's correlation analysis. Fisher's exact test was used in the contingency table analysis. Cumulative distributions were compared by Kolmogorov–Smirnov test. A single data point identified as outlier by ROUT (Robust regression and Outlier removal, $Q = 1\%$) in Fig. 6d ($y = 4.8$) was removed.

Data were reported as mean ± S.E.M., with $n$ indicating the number of neurons or animals, unless otherwise stated. Values of $P < 0.05$ were considered statistically significant.

**Neuronal model**. In Fig. 1d, simulations of Cl⁻ diffusion were performed with three dimensional finite elements modeling in the COMSOL Multiphysics environment. The diffusion coefficient for Cl⁻ in intracellular medium and in the pipette was $2.03 \cdot 10^{-9}$ m² s⁻¹ (see ref. [100]). Cl⁻-extrusion capacity was assumed to be uniformly distributed on the neuron membrane and was expressed in μmol m⁻² s⁻¹. Cl⁻ flux due to extrusion by KCC2 ($J_{KCC2}$) was also expressed in μmol m⁻² s⁻¹ and modeled with the following equation:

$$J_{KCC2} = V_{max} \frac{([Cl^-]_i - 3)/12}{1 + ([Cl^-]_i - 3)/12} \qquad (1)$$

where $V_{max}$ stands for maximal extrusion capacity, 3 mM is the Cl⁻ concentration at which Cl⁻ extrusion by KCC2 is null and 15 mM is the Cl⁻ concentration at which Cl⁻ extrusion is half maximal[56]. The soma of LI and LII neurons were modeled as ellipsoids with length of semi axes of 5.5 and 10.75 μm for LI neurons, as well as 4.58 and 7.9 μm for LII. Semi axes lengths were taken as the average of measurements performed in nine neurons in LI and nine neurons in LII. Dendrites were modeled as two cylinders of 1.5 μm diameter and 80 μm length originating from each pole of the soma. The extent of the dendritic tree beyond 80 μm was shown to have no significant effect on somatic [Cl⁻]ᵢ. The pipette was modeled as a cone, the tip had a diameter of 1.2 μm. The truncated cone had a length of 120 μm and a larger end of 22 μm diameter in accordance with our optical microscopy images of pipette tips. Simulations were performed until convergence of Cl⁻ concentration and Cl⁻ concentration was integrated in the soma to obtain a mean value. Simulations were performed for various values of Cl⁻-extrusion capacity ($V_{max}$) to obtain the curves in Fig. 2g.

The simulations displayed in Fig. 1g, h were derived as follows. To replicate the Cl⁻ load, we modeled a constant and uniformly distributed GABA-mediated Cl⁻ permeability. We computed the time course of net GABAergic current for a membrane potential held constant at –70 mV. The net GABAergic current is the sum of two opposing currents, a hyperpolarizing Cl⁻ current and a depolarizing bicarbonate (HCO₃⁻) current. Both currents were computed by the GHK flux equation. The Cl⁻ current is given by

$$I_{Cl} = P_{Cl} \frac{V_m F^2}{RT} \frac{[Cl^-]_i - [Cl^-]_o \exp\left(\frac{V_m F}{RT}\right)}{1 - \exp\left(\frac{V_m F}{RT}\right)}, \tag{2}$$

while the HCO₃⁻ current is given by

$$I_{HCO3} = P_{HCO3} \frac{V_m F^2}{RT} \frac{[HCO_3^-]_i - [HCO_3^-]_o \exp\left(\frac{V_m F}{RT}\right)}{1 - \exp\left(\frac{V_m F}{RT}\right)}. \tag{3}$$

The HCO₃⁻ permeability ($P_{HCO3}$) was set to one quarter of chloride permeability ($P_{HCO3} = 0.25\,P_{Cl}$)[56]. The Cl⁻ permeability ($P_{Cl}$) was adjusted so that the initial net current was equal to 10 pA in control condition, thus replicating the experimental results of Fig. 1f. The extracellular Cl⁻ concentration was set to 120 mM while the intracellular and extracellular HCO₃⁻ concentrations were set to 15 mM and 30 mM, respectively[56]. As above, Cl⁻ extrusion was obtained from Eq. (1). We used the estimated extrusion capacity for the control condition, while we set it to zero in furosemide condition. The dynamics of intracellular Cl⁻ concentration were given by

$$\frac{d[Cl^-]_i}{dt} = \frac{I_{Cl}}{F \cdot Vol} - Surf \cdot \frac{J_{Cl}}{Vol}. \tag{4}$$

where $F$ is the Faraday constant, Vol is the cell volume and $Surf$ the cell surface. Finally, the value of Cl⁻ reversal potential ($E_{Cl}$) was computed from the Cl⁻ concentration according to the Nernst equation

$$E_{Cl} = \frac{R \cdot T}{F} \log\left(\frac{[Cl^-]_i}{[Cl^-]_o}\right) \tag{5}$$

where $T$ stands for the absolute temperature and $R$ for the perfect gas constant.

For the simulations used to generate the results shown in Fig. 3d, we computed the steady-state Cl⁻ concentration resulting from the balance between synaptic Cl⁻ influx and Cl⁻ extrusion through KCC2. We used a single compartment model so all the synaptic inputs are considered to arrive at the soma. We first generated random trains of excitatory and inhibitory synaptic events with Poisson processes. Several values for the mean frequency of inhibitory events ($f_{inh}$) and for the mean frequency of excitatory events ($f_{exc}$) in the range 0–50 Hz were fed into Poisson processes to generate the random trains of synaptic events.

For the sake of simplicity, for both inhibitory and excitatory events, we assumed an instantaneous rise and an exponential decay of the synaptic conductance. Explicitly, synaptic conductances at time point $t_{j+1}$ were given by

$$g_{inh}\left(t_{j+1}\right) = g_{inh}\left(t_j\right)\left(1 - \frac{\Delta t}{\tau_{inh}}\right) + g_{unit,inh} \cdot \delta_{inh}\left(t_{j+1}\right), \tag{6}$$

$$g_{exc}\left(t_{j+1}\right) = g_{exc}\left(t_j\right)\left(1 - \frac{\Delta t}{\tau_{exc}}\right) + g_{unit,exc} \cdot \delta_{exc}\left(t_{j+1}\right) \tag{7}$$

where $\Delta t$ is the computational time step, $\tau_{inh} = 25$ ms[49] and $\tau_{exc} = 15$ ms[101] are the time constants of the inhibitory and the excitatory events, respectively. Moreover, $g_{unit,inh} = 1$ nS[102] and $g_{unit,exc} = 0.5$ nS[103] stand for the maximal conductance of a single inhibitory and excitatory event, respectively. Finally, $\delta_{inh}(t_{j+1})$ is equal to 1 if an inhibitory event is initiated at time step $j + 1$ and to 0 otherwise. The function $\delta_{exc}(t_{j+1})$ is defined analogously. The value of $P_{Cl}$ was inferred from the conductance to obtain the same current value under test conditions $V_m = –60$ and $[Cl^-]_i = 5$ mM. Cl⁻ and HCO₃⁻ currents were then computed from Eqs. (2) and (3), respectively.

High levels of synaptic activity (especially excitatory) as occur under capsaicin application are likely to trigger high-frequency spiking. This can depolarize the mean value of the membrane potential and increase the driving force of Cl⁻ currents, which might contribute to an increase of $[Cl^-]_i$. In our simulations, we thus added spiking mechanisms that were described by a Morris-Lecar model. This model was chosen for its simplicity and for the fact that it is conductance based. The sodium (Na⁺) and potassium (K⁺) currents through voltage gated channels were, respectively, given by

$$g_{Na} M(V)(V - E_{Na}) \text{ and } g_K W(V - E_K) \tag{8}$$

where the function $M(V)$ describes the fraction of open Na⁺ channels and is given by

$$M(V) = \frac{1 + \tanh\left(\frac{V - V_1}{V_2}\right)}{2}. \tag{9}$$

The variable $W$ describe the fraction of open voltage gated K⁺ channels at a given time and its dynamics is given by

$$\frac{dW}{dt} = \frac{W_{eq}(V) - W}{\tau_W(V)} \tag{10}$$

where

$$W_{eq}(V) = \frac{1 + \tanh\left(\frac{V - V_3}{V_4}\right)}{2} \tag{11}$$

and

$$\tau_W(V) = sech\left(\frac{V - V_3}{2V_4}\right). \tag{12}$$

The parameters $g_{Na}$ and $g_K$ stand, respectively, for the maximal conductances of voltage gated Na⁺ and K⁺ channels. For the functions related the Morris-Lecar spiking mechanisms, we use the following constants: $V_1 = –40$ mV, $V_2 = 18$ mV, $V_3 = –30$ mV, $V_4 = –30$ mV, $g_{Na} = 2$ nS and $g_K = 8$ nS. These parameters were chosen to obtain realistic time courses of membrane potential as displayed in supplementary Fig. 2c.

Significant spike rate is also likely to cause an accumulation of K⁺ in the extracellular space, which may mitigate the efficacy of Cl⁻ extrusion by KCC2. The dynamics of extracellular K⁺ in our model was described by the following equation[104]

$$\frac{d[K^+]_o}{dt} = \frac{I_K}{F \cdot (Vol/5)} + \frac{\left(K_{rest}^+ - [K^+]_o\right)}{\tau_K}. \tag{13}$$

Here, we assume that the volume of the extracellular space is one fifth of the intracellular volume. In this equation $\tau_K = 200$ ms stands for the time constant of K⁺ clearance in the extracellular space and $K_{rest}^+ = 3$ mM stands for the resting extracellular K⁺ concentration[104]. The term $I_K$ stands for the total K⁺ current, which is the net sum of the K⁺ leak current, the K⁺ current through voltage activated channels, the K⁺ current through the Na⁺-K⁺ ATPase pump and a term $J_{KCC2}/F$, which corresponds to the flux of K⁺ through KCC2 converted into current.

The total leak conductance was set to 1.2 nS and the leak K⁺ conductance was assumed to account for 80% of this conductance. The total leak current and the K⁺ leak current were thus, respectively

$$I_L = g_L(V - E_L) \text{ and } I_{LK} = g_{LK}(V - E_K) \tag{14}$$

where $E_L = –65$ mV is the leak potential, or resting membrane potential (RMP), and $E_K$ is the K⁺ equilibrium potential computed from the Nernst equation

$$E_K = \frac{R \cdot T}{F} \log\left(\frac{[K^+]_o}{[K^+]_i}\right) \tag{15}$$

with $[K^+]_i$ assumed constant at 120 mM. Since the activity of the Na-K ATPase pump plays a significant role in setting the extracellular K⁺ concentration, we also included it in our model. The K⁺ current through the pump was assumed constant and equal to 1.5 times the leak K⁺ current in order to maintain $[K^+]_o$ near 3 mM under resting conditions.

For this set of simulations, the equation describing the joint efflux of Cl⁻ and K⁺ through KCC2 (1) is modified to account for the possibility of variable extracellular K⁺ concentration, yielding

$$J_{KCC2} = V_{max} \frac{\left([Cl^-]_i - [K^+]_o\right)/12}{1 + \left([Cl^-]_i - [K^+]_o\right)/12}. \tag{16}$$

The total simulated time was 5000 s and we used a computational time step of 0.1 ms. The simulations were performed with the MATLAB software. To generate Fig. 3d, we took the average Cl⁻ concentration over the last third of the simulation. This was done to allow initial convergence of ionic concentrations.

The same model was used for simulating the dynamical effect of capsaicin as shown in supplementary Fig. 2c. For this figure, we assumed that according to internal data, the frequency of inhibitory and excitatory events in resting condition was 1 Hz in both instances. Under the effects of capsaicin, these frequencies climb to 5 Hz for inhibitory events and 50 Hz for excitatory ones also according to internal data. To best replicate experimental results, we assumed that the frequency of synaptic events increased linearly from resting frequencies to capsaicin frequencies in the time span of 2 min.

**Reporting summary**. Further information on research design is available in the Nature Research Reporting Summary linked to this article.

## Data availability

All data supporting the findings of this study are available within the article or from the corresponding author upon reasonable request. Source data are provided with this paper.

## Code availability

Codes and software are available upon request. Source data are provided with this paper.

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

## Acknowledgements

This work was supported by the People Program (Marie Curie Actions) of the European Union's Seventh Framework Program (FP7/2007-2013) under REA grant agreement no. 318 997—NEUREN (F.F. and Y.D.K.), the Regione Piemonte/University of Turin Fellowship Program (FF), the Canadian Institute of Health Research (grant MOP 12942 and FDN 159906 to Y.D.K.), a Mexican Science Foundation scholarship (CONACYT #312229 to J.P.S.) and the Canada Research Chair program (Y.D.K). A.G.G. is a Chercheur-Boursier (Fonds de recherche du Québec–Santé), and was supported by a Sentinel North Partnership Research Chair and grant #06507 from the Natural Sciences and Engineering Research Council of Canada. We thank Mr. Sylvain Côté for expert assistance with artwork.

## Author contributions

F.F., J.P.-S., and Y.D.K. conceived and designed the project. F.F., A.M., Y.D.K. supervised the experiments. F.F., J.P.-S., S.F., L.-E.L., I.P.-F., M.C., F.W., and C.S. performed the experiments. N.D. performed computer simulations and contributed to interpretation of results. A.G.G. and L.-E.L. developed image analysis methods. F.F., J.P.-S., L.-E.L., I.P.-F., M.C., A.C., S.F., and A.G.G. analyzed the data. F.F., J.P.-S., and Y.D.K. wrote the manuscript. All the authors read and discussed the manuscript.

## Competing interests

The authors declare no competing interests.

**Additional information**

