## [Peer Review File · Nature Communications]

Reviewers' Comments:

Reviewer #1:

Remarks to the Author:

Noxious stimulation can induce a lasting sensitization of nociceptive neurons within the spinal cord dorsal horn, an alteration that is thought to contribute to the development of chronic pain. Recent work has linked the development of this central sensitization to the release of the neurotrophin BDNF and the down-regulation of the Cl⁻ co-transporter KCC2. The latter brings about an increase in intracellular Cl⁻ concentrations, which attenuates the hyperpolarizing (inhibitory) effect of GABA. This ionic plasticity is thought to enable the development of central sensitization.

Ferrini et al. used electrophysiological techniques, supplemented with cellular imaging, immunohistochemistry, and modeling, to assess KCC2 activity within lamina I and II of the superficial dorsal horn. These lamina can be distinguished on the basis of their afferent input, which can be visualized with staining for neurokinin (NK1; lamina I) and isolectin B4 (IB4; lamina II). Functionally, enhanced thermal pain has been linked to alterations in lamina I while increased pain to mechanical stimulation is tied to neurons within inner lamina II and the deep dorsal horn. The authors show that laminae I and II also differ in the extent to which they exhibit activity-dependent ionic plasticity. This difference was linked to a lower density of KCC2 within lamina I and greater expression of the BDNF receptor (TrkB). Evidence is presented that this reduces the GABA-dependent brake on neural excitation in lamina I, which allows for runaway LTP—relative to lamina II, where LTP is more restrained. The results have a number of important implications. First, they reveal a process that would enable the development of thermal hyperalgesia. Second, the work clarifies why pathological conditions that induce KCC2 hypofunction (e.g., spinal cord injury) fuel maladaptive plasticity and mechanical pain. The work also represents an important scientific advance, providing electrophysiological evidence that ionic plasticity can have a metaplastic effect.

In general, the methods and analyses were well executed and described. Further, the text and graphics are in excellent shape. In addition, the research was very thorough, relying on multiple methods to build a strong story. Finally, because the article has important implications regarding the regulation of plasticity within the central nervous system, I believe that it will be of general interest to researchers within the neuroscience community. The comments that I have focus on just a few issues that I thought needed some clarification/elaboration.

One issue that emerged a number of times within the paper involved the interpretation of findings under alternative conditions. For example, whether the MQAE effect reported in Figure 3c varies depending upon whether there was, or was not, synaptic activity. Another example concerned the data presented in Figure 6c, where it is suggested that ANA-12 has a significant effect in lamina I, but not lamina II. In cases such as these, the claim involves a form of interaction, which appears to have been inferred from the pattern of significance obtained (effect X was significant under condition A, but not condition B). Where possible, it would be helpful if the author's presented evidence that the interaction term (e.g., from an

ANOVA) was statistically significant. Another issue regarding their analyses stemmed from the data presented in Fig. 5d, where it appears that the two left-most data points were excluded. If that is true, why was this done?

There were a number of places where the drug treatment was changed. On the one hand, this bolsters the generality of the author's claims. It was not, however, always clear why a change was made. For example, furosemide was assessed in combination with bumetanide in Fig. 1d, but furosemide alone was tested in Fig. 1e. Later the author's switched to the KCC2 antagonist VU0240551, but this was not explained.

In the text, it would be helpful if the author's reminded the reader of the theoretically expected value on line 132. Later (lines 235-237) the authors suggest that the lower levels of KCC2 expression in lamina I appear linked to higher levels of TrkB activation. Bolstering this claim with some empirical observations would help. In the Discussion section the authors relate their findings to alterations in KCC2 function caused by pathology. Is there any reason to believe that the present results were affected by the methods used to isolate and record from the spinal cord tissue? Was there, for example, a shift over time? Could a loss in 5HT input affect the results? Would the loss of descending regulatory processes have different effects on lamina I and II function? On lines 325-330, they discuss how KCC2 hypofunction would foster mechanical allodynia. Would the relations discovered in the present study suggest that, in the absence of pathology, the system is prone to thermal hyperalgesia? Is there any evidence for this? And finally, the model illustrated in Figure 9 assumes a unidirectional effect of BDNF on KCC2. Yet, Rivera, the present authors, and evidence from our laboratory suggest a bidirectional effect, potentially linked to PLC-gamma. How might this alter the presumed relations and the consequences of injury? Of course, these questions raise a host of issues that go well beyond the present paper. Nonetheless, some commentary would be helpful.

James Grau

Reviewer #2:

Remarks to the Author:

This interesting study of the heterogeneity of KCC2 expression in the spinal cord is focused on differences in the superficial vs deep lamina of the dorsal horns. This heterogeneity in chloride transport capacity is novel and the potential for cell-specific activity-dependent modulation of GABA signaling is an important new insight. Overall the paper is clearly written and the experimental logic is sound, but there are many areas where the strength of evidence could be readily improved. The LTP / TrkB element of this study is the weakest. Although additional experiments could be performed that might bolster these findings, the story is equally compelling without this section. The functional impact of Cl transport rate is also clear from the last section of the Discussion.

1. Figure 1:

- a. The point of the paper is that lamina 1 has a different Cl transport capacity than lamina 2. Why then are data from lamina 1 and 2 lumped together to obtain $E_{Cl} = -47.9 \pm 0.9$

mV for the entire superficial dorsal horn? If transport capacity were different in the 2 lamina, then ECl should be different in the loaded condition for lamina 1 neurons vs. lamina 2 neurons.

b. Why is the range of ECl across 72 cells from the 2 lamina so small (-47.9 +/- 0.9 mV) yet when the next group of 6 cells is subject to the same assay (the control group for the transport inhibition, figure 1d), the latter group has an ECl of -43.8 and a std dev of 2.8 mV? A two-tailed T test indicates that the chance that these neurons came from the same population is essentially 0 ($< 1 \times 10^{-25}$)! This does not engender much confidence in the reader, particularly since the point of this study is measurement of population variance.

c. Figure 1e: this experiment is supposed to show heterogeneity in transport rates. Yet a trivial Cl flux is used to provoke transport. The Cl flux is small because the driving force is only the difference between native ECl and RMP; i.e. a driving force of about 10 mV. Accordingly, the evoked currents are tiny – only 10 pA at the peak, with rapid reduction thereafter. A more effective strategy would be to step the membrane potential far from the predicted ECl, evoke a large inward Cl flux, and then step back to -70 mV and test for the size and direction of the GABA-activated current, e.g. Staley and Proctor J Physiol. 1999 fig 3.

i. Why aren't individual evoked currents resolvable with each GABA puff in Fig 1e?

ii. How is desensitization separated from shift in GABA reversal?

iii. It is very difficult to understand how somatic Cl flux from the 10pA currents shown in the top 2 panels can drive ECl to the holding potential in the absence of transport (lower panel, solid line): the somatic volume would need to be unrealistically small to obtain such complete dialysis by such a small Cl flux.

2. Figure 3:

a. Panel b:

i. what is the holding potential and electrode fill? What is the expected ECl?

ii. Please use a fill and test potential that permits resolution of glutamate vs GABAA postsynaptic currents.

iii. The frequency of these currents are not specified, but appear to be approximately 10 Hz. There does not appear to be a measurable shift in ECl as assessed from the size of the sIPSCs, but the consequence of this shift can't be predicted without knowing the holding potential, electrode fill, i.e. driving force for Cl influx.

iv. Clarification of depolarizing activity (e.g. glutamatergic PSCs) are important in panel b because there is otherwise insufficient driving force to change ECl in panel C. Excessive GABAA currents will drive ECl to RMP, but from Figure 2f we know that this is a very small driving force.

b. Panel c:

i. to what cytoplasmic Cl do MQAE lifetimes of 4.0 vs 4.5 ns correspond? Without this information, it is not possible to assess the importance of this panel

ii. were the starred t-tests corrected for multiple comparisons?

iii. Most of the differences in high vs low synaptic activity arise from 2 measures when synaptic activity is blocked at 20-30 um from the border. The standard deviation is quite high for these measures. These are imaging experiments so additional cells could be added to reduce variance at these key measures. It would also be helpful to let the reader know how many cells were assayed for each point.

c. Panel d: there is not sufficient information to understand this panel. Where are the IPSCs

arriving? Soma only? Soma + dendrites? What is RMP – this should be drawn on the panel.

3. Figure 4

a. Panels a-c: As stated in the introduction, the HCO₃ permeability is much less than Cl permeability through the GABA_A conductance. The evoked currents are proportionately smaller. It is not compelling to compare an HCO₃ current that is 4x smaller than the Cl current to demonstrate stability of the HCO₃ current. The evoked currents need to be of similar size for this evidence to bear weight. C.F. Figure 5 of Staley and Proctor 1999.

4. Figure 5

a. IHC is not a compelling method to demonstrate differences in protein

b. The IHC needs to have a control that corresponds to the total neuronal membrane area in each lamina. Perhaps a microtubule stain?

c. Panel d: the starred p value appears to apply to the last 5 measures only? Is this a measure of probability that the slope is nonzero? Why are only the last 5 points included?

How many repeated measures were used – would need to account for all possible subgroups of 5 contiguous points.

5. Figure 7

a. The argument that KCC2 is affecting disinhibition at 2Hz is not congruent with the predictions of figure 3d, which indicates that ECI is stable at that PSC frequency.

b. 2 Hz Cl loading is also not congruent with Figure 4a-c where Hz stimuli at 10x higher frequency were used to demonstrate Cl loading.

c. VU0240551 has substantial nonspecific effects at 10 uM. These experiments must be repeated with GABA receptors blocked to assess the nonspecific VU024 effects in this preparation.

d. Similarly, BDNF TrkB receptors have many effects beyond KCC2 expression, and repeating the experiments with GABA_A receptors blocked are critical.

Discussion:

Importance of KCC2 vs Donnan and effects of synaptic Cl loading: Please note that the variance in ECI does not decrease when KCC2 is blocked in lamina 1 in Figure 2C. If variance in KCC2 activity accounted for the variance in ECI in Cl-loaded neurons, then blocking KCC2 should have removed this variance. The stable variance in control vs furosemide argues quite strongly for the views expressed in reference 30. So does the variance in ECI shown in figure 2f. This is not to discount the importance of the finding of variance in KCC2 activity, which is another layer of complexity in GABA signaling.

Kevin Staley

Reviewer #3:

Remarks to the Author:

The manuscript by Ferrini et al. describes experiments revealing a difference in distribution of the chloride transporter, KCC2, in the superficial laminae of the dorsal horn. The authors go on to demonstrate that in lamina I (a region that includes key brain-projecting neurons) under conditions of chloride stress and high synaptic activity, KCC2 can no longer keep up with the chloride accumulation and this leads to a functional decrease in inhibition to the region. Lamina II, a heterogeneous group of interneurons, have a higher expression of KCC2 and less effect on inhibition during a chloride load. The authors drive synaptic release

of glutamate in vitro using capsaicin and in vivo using dorsal root stimulation, and find that apparent excitatory transmission in lamina I is enhanced considerably more than lamina II. They suggest that their work presents an important new clue to how synaptic activation can promote hyperalgesia by increasing synaptic drive via lamina I neurons. Overall the work is carefully and well done and the project is of broad interest. However, there is a connection missing between their in vitro work and in vivo work that makes the results less convincing.

Suggestions for revision:

1. The authors have suggested an intriguing interaction between synaptic activation and a failure of inhibition to lamina I neurons. However, there remains an important missing link in the manuscript in its present form. The in vivo field potential recordings are a strength in terms of what may occur under physiological conditions. By contrast, the interpretation is more tenuous. To connect the clean slice recording data of direct measurements of ECI in individual neurons and spontaneous capsaicin-driven IPSCs with the in vivo field potentials during dorsal root stimulation, it would be important to add an experiment in the slice stimulating evoked EPSCs before and after 2Hz stimulation, and preferably repeating the result with the trkB antagonist. Without this experiment, the link between the in vivo pre and post 2Hz stimulation data and the in vitro differences in ECI is missing. Field potentials are inherently complex (see below), and blocking trkB receptors for several hours will have multiple effects on the circuit at many levels other than ECI in the lamina I and II neurons.
2. Because TrkB inhibition could have multiple effects in the circuit other than direct effects on extrusion of Cl⁻, the TrkB inhibitor ANA should be used to show that four hours after in vivo injection, there is a significant difference in the rundown of eIPSCs during a train in the slice, as shown in figure 4a.
3. The measurement of the area of a field potential in vivo is misleading. LTP is commonly used to refer to synaptic changes in the monosynaptic EPSC/P, not to the multiple polysynaptic events that follow the initial EPSP in this circuit. While it is difficult to distinguish the two in field potential recordings, the common approach to get around this has been to use field EPSP slope, as this is arguably dominated by earlier events (initial primary afferent EPSPs). Measuring an area over hundreds of milliseconds is not unreasonable for the arguments in this manuscript, including as it does many recurrent synapses, since the loss of inhibition will enhance any EPSC on lamina I cells regardless of whether they originate in the dorsal root or from central neurons. For the main point of this paper, this approach is acceptable as long as the authors make clear that this should not be thought of as LTP in the conventional sense.
4. The "runaway" rise of the field potential area may or may not have anything to do with LTP at the primary afferent synapse, and should be called something else. Moreover, it is notable that the deeper layer field potentials also increase continuously over hours, and that even after treatment with ANA, both deep and superficial field potentials increase over time (albeit at a dramatically shallower slope) is important as well, suggesting that this may be typical of this set of synapses.

Minor:

1. It would be worth pointing out that lamina II neurons (as well as lamina I neurons) are a highly heterogeneous population of cells. Do the immune data suggest that ALL neurons in

each lamina show differences in KCC2? A good discussion of this point would be of use to those in the field.

Reviewer #1 (Remarks to the Author):

Noxious stimulation can induce a lasting sensitization of nociceptive neurons within the spinal cord dorsal horn, an alteration that is thought to contribute to the development of chronic pain. Recent work has linked the development of this central sensitization to the release of the neurotrophin BDNF and the down-regulation of the Cl⁻ co-transporter KCC2. The latter brings about an increase in intracellular Cl⁻ concentrations, which attenuates the hyperpolarizing (inhibitory) effect of GABA. This ionic plasticity is thought to enable the development of central sensitization.

Ferrini et al. used electrophysiological techniques, supplemented with cellular imaging, immunohistochemistry, and modeling, to assess KCC2 activity within lamina I and II of the superficial dorsal horn. These lamina can be distinguished on the basis of their afferent input, which can be visualized with staining for neurokinin (NK1; lamina I) and isolectin B4 (IB4; lamina II). Functionally, enhanced thermal pain has been linked to alterations in lamina I while increased pain to mechanical stimulation is tied to neurons within inner lamina II and the deep dorsal horn. The authors show that laminae I and II also differ in the extent to which they exhibit activity-dependent ionic plasticity. This difference was linked to a lower density of KCC2 within lamina I and greater expression of the BDNF receptor (TrkB). Evidence is presented that this reduces the GABA-dependent brake on neural excitation in lamina I, which allows for runaway LTP—relative to lamina II, where LTP is more restrained. The results have a number of important implications. First, they reveal a process that would enable the development of thermal hyperalgesia. Second, the work clarifies why pathological conditions that induce KCC2 hypofunction (e.g., spinal cord injury) fuel maladaptive plasticity and mechanical pain. The work also represents an important scientific advance, providing electrophysiological evidence that ionic plasticity can have a metaplastic effect.

In general, the methods and analyses were well executed and described. Further, the text and graphics are in excellent shape. In addition, the research was very thorough, relying on multiple methods to build a strong story. Finally, because the article has important implications regarding the regulation of plasticity within the central nervous system, I believe that it will be of general interest to researchers within the neuroscience community. The comments that I have focus on just a few issues that I thought needed some clarification/elaboration.

We wish to thank reviewer#1 for the positive comments regarding this study.

One issue that emerged a number of times within the paper involved the interpretation of findings under alternative conditions. For example, whether the MQAE effect reported in Figure 3c varies depending upon whether there was, or was not, synaptic activity. Another example concerned the data presented in Figure 6c, where it is suggested that ANA-12 has a significant effect in lamina I, but not lamina II. In cases such as these, the claim involves a form of interaction, which appears to have been inferred from the pattern of significance obtained (effect X was significant under condition A, but not condition B). Where possible, it would be helpful if the author's presented evidence that the interaction term (e.g., from an ANOVA) was statistically significant. Another issue regarding their analyses stemmed from the data presented in Fig. 5d, where it appears that the two left-most data points were excluded. If that is true, why was this done?

As suggested, we now added, where applicable, multiple comparisons between groups using both one-way or two-way ANOVAs as well as regression analyses. These apply to Figures 3 to 8.

-In the case of Fig. 3c (MQAE imaging), the experiment was conceived to compare two extreme scenarios (no synaptic activity with high synaptic activity) to unveil activity-dependent Cl⁻ accumulation. The new analysis by linear regression confirmed significance change in [Cl⁻]_i as a function of depth. A significant effect of synaptic activity on [Cl⁻]_i was also confirmed by two-way ANOVA.

-Concerning former-Fig. 6c (now Fig. 5c), Two-way ANOVA analysis confirms the difference in the KCC2 expression according to the laminar position and demonstrates a significant interaction between lamina and treatment.

Concerning former Fig. 5d (now Fig. 4d), we agree that the figure presentation was misleading: as they gave the wrong impression that different ranges of depth were analyzed. We have now limited the deeper limit of lamina II to 80 μm for all graphs and quantification to be homogeneous across all analyses. In the specific case of Fig 4d, linear regression analysis confirms the existence of a gradient with depth. In addition, a one-way ANOVA confirmed difference in KCC2 intensity across depth.

There were a number of places where the drug treatment was changed. On the one hand, this bolsters the generality of the author's claims. It was not, however, always clear why a change was made. For example, furosemide was assessed in combination with bumetanide in Fig. 1d, but furosemide alone was tested in Fig. 1e. Later the author's switched to the KCC2 antagonist VU0240551, but this was not explained.

We agree that changing drug treatment may have generated confusion to the reader. In the first pharmacological experiment (former Fig. 1d, now 1e) focusing on single cell responses to GABA applications, we tested the effect of bumetanide to confirm the lack of NKCC1 contribution in adult dorsal horn neurons. This is why in following experiments, using GABA applications on DH neurons (former Fig. 1e, now 1f; Fig. 2c and new Supplementary Fig. 1), we used furosemide alone since bumetanide had no effect in these conditions. For experiments focusing on synaptic responses to electrical stimulation of afferents (Fig. 7), we had to avoid confounding effects of drugs acting on NKCC1 since NKCC1 is highly expressed in primary afferents while KCC2 is not. This is why we reverted to VU0240551 as a more specific blocker of KCC2 than furosemide. Yet, as reviewer 2 raised concerns on the specificity of VU0240551 (see response to Reviewer 2), and to ensure consistency across drugs, we repeated the latter experiment in the continuous presence of bumetanide to subtract any effect of furosemide on NKCC1 (New Supplementary Fig. 7a). We now clarified this on page 9 of the manuscript.

In the text, it would be helpful if the author's reminded the reader of the theoretically expected value on line 132.

As suggested, we indeed added the theoretical (calculated) value in brackets (-37mV).

Later (lines 235-237) the authors suggest that the lower levels of KCC2 expression in lamina I appear linked to higher levels of TrkB activation. Bolstering this claim with some empirical observations would help.

To address this reviewer's suggestion, as well as that of reviewer #3, we have now added additional experiments linking TrkB signaling to KCC2 function. These are reported in new Fig. 6d and new supplementary Fig. 4a). Briefly, ANA-12 pre-treatment prevents activity-dependent Cl^- accumulation induced by repetitive synaptic activity in lamina I, while has little effect in lamina II, indicating that blocking TrkB enhances Cl^- extrusion capacity in lamina I. In addition, we now also show that, applying a specific KCC2 enhancer (CLP257) has the same effect as ANA-12 in this paradigm (new Fig. 6d). The same parallel is now also demonstrated in the synaptic facilitation paradigm (Fig. 7): the KCC2 enhancer, similarly to ANA-12 constrains runaway synaptic facilitation in the superficial laminae (new Fig. 7g and new supplementary Fig. e). CLP257 was used at 5 μM , below any potential side effects, as previously shown (Lorenzo et al *Nat Commun* 2020; Ostroumov et al *Neuron* 2016; see also Gagnon et al *Nat Med* 2013 and Gagnon et al *Nat Med* 2017) These three additional sets of experiments thus support the hypothesis that on-going TrkB signaling downregulates KCC2 activity in the superficial dorsal horn.

In the Discussion section the authors relate their findings to alterations in KCC2 function caused by pathology. Is there any reason to believe that the present results were affected by the methods used to isolate and record from the spinal cord tissue? Was there, for example, a shift over time? Could a loss in 5HT input affect the results? Would the loss of descending regulatory processes have different effects on lamina I and II function?

Shift over time. We do not think that this may have occurred, because: to minimize possible alterations due to ongoing changes in slices over time after dissection, slices exposed to different treatments were randomly selected for subsequent recording; similarly, neurons in slices were randomly recorded in lamina I or in lamina II, without following a systematic order.

Spinal extraction approach. We do not believe the results are affected by the spinal isolation methods because a series of *in vivo* experiments confirm the *ex vivo* results. First, all of the KCC2 quantification from immunostaining was performed from animals perfused before the spinal cord was extracted. Second, the finding that ANA-12 raises KCC2 expression in lamina I resulted from *in vivo* experiments (Fig. 5a-d). Third, we now report results from *in vivo* optogenetics experiments showing that greater sensitization by activation of superficially projecting afferents is prevented by treatment with the KCC2 enhancer CLP290 (new Fig. 8g). Fourth, the differential impact of spinal inhibitory transmission on modality-specific pain sensitivity (heat vs. mechanical) is consistent with weaker inhibition in the superficial dorsal horn (new Fig. 8i,j).

The reviewer's reference to the report by Bos et al., (2013) that 5HT-mediated transmission may enhance KCC2 is pertinent. However, for the reasons outlined above we do not think loss of descending 5HT input in spinal explants could explain the finding of differential KCC2 expression across laminae. We have added a statement in discussion to address this.

On lines 325-330, they discuss how KCC2 hypofunction would foster mechanical allodynia. Would the relations discovered in the present study suggest that, in the absence of pathology, the system is prone to thermal hyperalgesia? Is there any evidence for this?

To directly address this important point, we performed additional experiments to assess the impact of KCC2 hypofunction on modality-specific sensitization. In new Fig. 8 and new suppl. Fig 8, we show that sensitization from optogenetic activation of TRPV1 primary afferents is larger and longer than that from optogenetically activated MRGPRD afferent fibers. To test for causal involvement of KCC2 in this differential sensitization, we treated mice with the KCC2 enhancer CLP290. CLP290 is the CLP257 carbamate prodrug, designed to protect the hydroxyl group from glucuronidation which improves its bioavailability (Gagnon et al *Nat Med* 2013). Systemic (oral) administration of CLP290 was previously shown to increase KCC2 expression in the CNS (Ferrini et al *Sci Rep* 2017, Chen et al *eLife* 2017; Lizhnyak et al *J Neurotrauma* 2019). We found that CLP290 had a greater effect on TRPV1 fiber-induced sensitization than that from MRGPRD afferents. Together, these data support the hypothesis that, in absence of pathology, the system is indeed more prone to thermal than mechanical hyperalgesia.

And finally, the model illustrated in Figure 9 assumes a unidirectional effect of BDNF on KCC2. Yet, Rivera, the present authors, and evidence from our laboratory suggest a bidirectional effect, potentially linked to PLC-gamma. How might this alter the presumed relations and the consequences of injury? Of course, these questions raise a host of issues that go well beyond the present paper. Nonetheless, some commentary would be helpful.

We agree with the reviewer that a number of findings indicated opposite direction in regulation of KCC2 by BDNF (e.g., immature vs. mature tissue, early vs later time points after injury, etc.). However, as pointed out by the reviewer, these issues go well beyond the scope of the present

study which does not address development nor response to injury. We have nevertheless added a comment to the discussion highlighting that our results indicate that in contrast to other situations, in normal, adult tissue, BDNF-TrkB signaling appears to negatively regulate KCC2. Beyond this, stay tuned for follow up papers!

Reviewer #2 (Remarks to the Author):

This interesting study of the heterogeneity of KCC2 expression in the spinal cord is focused on differences in the superficial vs deep lamina of the dorsal horns. This heterogeneity in chloride transport capacity is novel and the potential for cell-specific activity-dependent modulation of GABA signaling is an important new insight. Overall the paper is clearly written and the experimental logic is sound, but there are many areas where the strength of evidence could be readily improved. The LTP / TrkB element of this study is the weakest. Although additional experiments could be performed that might bolster these findings, the story is equally compelling without this section. The functional impact of Cl transport rate is also clear from the last section of the Discussion.

We thank the reviewer for finding our study interesting and the story compelling.

1. Figure 1:

a. The point of the paper is that lamina 1 has a different Cl transport capacity than lamina 2. Why then are data from lamina 1 and 2 lumped together to obtain $E_{Cl} = -47.9 \pm 0.9$ mV for the entire superficial dorsal horn? If transport capacity were different in the 2 lamina, then E_{Cl} should be different in the loaded condition for lamina 1 neurons vs. lamina 2 neurons.

The main purpose of Figure 1 is to first introduce and highlight the heterogeneity of Cl⁻ transport throughout the superficial dorsal horn (Lamina I + II). This observation has never been properly tested under Cl⁻ loading conditions. The finding is, in turn, what justifies the follow up experiments separating lamina I and II measurements. We realize this was not clear because it was not illustrated as such in Fig. 1. To make it clearer, we have now added a histogram illustrating the variance in [Cl⁻]_i across all recordings when measured under a patch pipette-imposed Cl⁻ load (new Fig. 1c).

b. Why is the range of E_{Cl} across 72 cells from the 2 lamina so small (-47.9 ± 0.9 mV) yet when the next group of 6 cells is subject to the same assay (the control group for the transport inhibition, figure 1d), the latter group has an E_{Cl} of -43.8 and a std dev of 2.8 mV? A two-tailed T test indicates that the chance that these neurons came from the same population is essentially 0 ($< 1 \times 10^{-25}$)! This does not engender much confidence in the reader, particularly since the point of this study is measurement of population variance.

We mistakenly used standard error the mean (SEM) for the large sample (72 cells), which is not justified when the n is so large. Standard deviation (SD) is more appropriate in such conditions. This is likely what misled the reviewer. To better illustrate the population variance in Cl⁻ extrusion capacity, as mentioned above, it is now presented in the form of a histogram superimposed by a Gaussian fit along with the standard deviation. Concerning the specific example highlighted by the reviewer, E_{GABA} from the overall sample of SDH neurons (n=72) is -48 mV ± 8 SD. From this, the control values in the experiment with NKCC1/KCC2 blockers the reviewer refers to (-44 mV ± 2.8 SEM or ± 6.5 SD) can be considered, statistically, to belong to the same population (P is 0.24).

c. Figure 1e: this experiment is supposed to show heterogeneity in transport rates. Yet a trivial Cl flux is used to provoke transport. The Cl flux is small because the driving force is only the difference between native E_{Cl} and RMP; i.e. a driving force of about 10 mV. Accordingly, the evoked currents are tiny – only 10 pA at the

peak, with rapid reduction thereafter. A more effective strategy would be to step the membrane potential far from the predicted E_{Cl} , evoke a large inward Cl flux, and then step back to -70 mV and test for the size and direction of the GABA-activated current, e.g. Staley and Proctor J Physiol. 1999 fig 3.

i. Why aren't individual evoked currents resolvable with each GABA puff in Fig 1e?

ii. How is desensitization separated from shift in GABA reversal?

iii. It is very difficult to understand how somatic Cl - flux from the 10pA currents shown in the top 2 panels can drive E_{Cl} to the holding potential in the absence of transport (lower panel, solid line): the somatic volume would need to be unrealistically small to obtain such complete dialysis by such a small Cl flux.

The purpose of this experiment is to show that near V_r , and under physiological $[Cl^-]_i$, during a sustained GABA input, the polarity of GABA_A current is maintained under normal condition, while it shifts from hyperpolarizing to depolarizing when KCC2 is blocked. In response to the reviewer's concern we should point out that, in these experiments, we are not measuring Cl^- flux, but net GABA_A current which, as the reviewer is well aware, consists of a HCO_3^- and a Cl^- mediated component. This was perhaps unclear in previous Fig. 1e which referred only to "current". To clarify this, we now plotted in the simulation graph in Fig 1g, the different ionic currents (Cl^- , and HCO_3^-) as well as the resulting net currents. From the new plot, it is clear that the magnitude of the Cl^- current is larger than 10 pA (near 30 pA). Moreover, it is sustained over time making the actual charge transfer more significant (~250 pC). These simulations also take into account the soma size of typically small dorsal horn neurons. To better illustrate the results of our simulations, we added a new panel Fig. 1h which shows that a sustained GABA_A current of ~10 pA, delivered when the neuron is at rest, induced a depolarization of about 8 mV on E_{Cl} when KCC2 is blocked, and this can realistically account for the experimentally observed change in polarity shown in former Fig. 1e (now 1f).

In response to the other reviewer's specific concerns:

i- the individual currents are not easily resolvable because the net GABA_A current is small (holding at -70 mV, very close to E_{GABA}). This, added to temporal summation of the typically slower GABA-evoked currents makes it hard to resolve them individually.

ii- desensitization cannot be separated by the shift in E_{GABA} in this experiment. This is addressed specifically in Fig. 6;

iii- this is now addressed by the fact that, as described above, the Cl^- current itself is > 10 pA (actually 30 pA) and that the dorsal horn neurons have indeed very small cell bodies. The new simulation illustrated in Fig. 1h takes these parameters into account and demonstrates consistency.

As suggested by the reviewer, we have also replicated the experiment by Staley and Proctor J Physiol. 1999-Fig. 3 (now illustrated in new Supplementary Fig. 1). This experiment allows to evaluate the rate of recovery of GABA_A currents recorded near the resting potential, following a large conditioning GABAergic pulse at depolarized holding voltage ($V_h = -10$ mV, in our protocol). The first GABA_A current after the conditioning pulse is depolarizing, but it quickly reverts to fully hyperpolarizing within 2 seconds. When furosemide is applied to the same neuron, the GABA_A current remains depolarizing for 3 seconds after the conditioning pulse and biphasic for an additional 2 seconds, indicating a strong delay in the recovery of GABA_A polarity and therefore confirming the role of KCC2 in fighting a Cl^- load.

2. Figure 3:

a. Panel b:

i. what is the holding potential and electrode fill? What is the expected E_{Cl} ?

ii. Please use a fill and test potential that permits resolution of glutamate vs GABA postsynaptic currents.

iii. The frequency of these currents are not specified, but appear to be approximately 10 Hz. There does not appear to be a measurable shift in ECI as assessed from the size of the sIPSCs, but the consequence of this shift can't be predicted without knowing the holding potential, electrode fill, i.e. driving force for Cl⁻ influx.
iv. Clarification of depolarizing activity (e.g. glutamatergic PSCs) are important in panel b because there is otherwise insufficient driving force to change ECI in panel C. Excessive GABAA currents will drive ECI to RMP, but from Figure 2f we know that this is a very small driving force.

We agree with the reviewer that the previous trace showing the overall increase in synaptic activity induced by capsaicin was confusing and did not allow to properly discriminate sEPSCs and sIPSCs. We have now added a set of experiments in which spontaneous postsynaptic currents are recorded with a Cs-methansulfonate based pipette solution either at -60 mV (to isolate sEPSCs) or at 0 mV (to isolate sIPSCs). Values of sIPSCs/sEPSCs frequencies and amplitudes are now reported in new Supplementary Fig. 3a-d. The purpose of this experiment was only to show (as demonstrated in past studies) that capsaicin efficiently increases synaptic activity in the dorsal horn (analysis of the collapse in IPSCs is examined systematically in Fig. 6). But the reviewer is right that taking into account increase in excitatory activity by capsaicin is critical because it drives depolarization, enhancing the driving force for Cl⁻ influx. We, in fact, found that the frequency of EPSCs in capsaicin is about 10x higher than that of IPSCs (as expected since it acts on primary afferent terminals). Thus, the expected increase in intracellular Cl⁻ will result from a combination of: *i)* an increase in inhibitory input; *ii)* an increase in driving force due to (depolarizing) excitatory transmission and spiking activity (new supplementary Fig 2b-c); *iii)* reduction of KCC2 extrusion due to firing-induced increased extracellular K⁺ (Doyon et al *Neuron* 2016). The simulations illustrated in the new version of Fig. 3d now take also into account the effect of excitation.

b. Panel c:

i. to what cytoplasmic Cl do MQAE lifetimes of 4.0 vs 4.5 ns correspond? Without this information, it is not possible to assess the importance of this panel

ii. were the starred t-tests corrected for multiple comparisons?

iii. Most of the differences in high vs low synaptic activity arise from 2 measures when synaptic activity is blocked at 20-30 um from the border. The standard deviation is quite high for these measures. These are imaging experiments so additional cells could be added to reduce variance at these key measures. It would also be helpful to let the reader know how many cells were assayed for each point.

i. The fluorescence lifetimes reported in the original version had a systematic offset that was now corrected. To address the reviewer's question, we have also converted them into Cl⁻ concentration based on calibrations as described in Doyon et al *Plos Comput Biol* 2011.

ii. We now applied a more thorough statistical analysis of the relationship between [Cl⁻]_i and distance: a two-way ANOVA revealed significant difference due to the treatment and regression analysis that the slopes are different. This is now outlined in Results and the figure legend.

iii. In the revised Fig. 3c, the data points now include larger Ns and the Ns are specified.

c. Panel d: there is not sufficient information to understand this panel. Where are the IPSCs arriving? Soma only? Soma + dendrites? What is RMP – this should be drawn on the panel.

As described above, the *in silico* simulations in revised Fig. 3d were conducted to take into account excitation. We now used [Cl⁻]_i in the Y axis for more direct comparison with Fig 3c. We used a single compartment model so all the IPSCs and EPSCs are considered to arrive at the soma. This has been clarified in the Figure legend. The RMP was defined in our model by the reversal

potential of leak channels and was set to -65 mV. We now included more detailed information on the model and the parameters used for the simulation in the Methods section.

3. Figure 4

a. Panels a-c: As stated in the introduction, the HCO₃⁻ permeability is much less than Cl⁻ permeability through the GABA_A conductance. The evoked currents are proportionately smaller. It is not compelling to compare an HCO₃⁻ current that is 4x smaller than the Cl⁻ current to demonstrate stability of the HCO₃⁻ current. The evoked currents need to be of similar size for this evidence to bear weight. C.F. Figure 5 of Staley and Proctor 1999.

We did not explain properly in the original version the purpose of the comparison between the recordings at 0mV vs -90mV. The point of the experiment was indeed not to compare a collapse in Cl⁻ gradient to that of a HCO₃⁻ gradient but rather the collapse of the Cl⁻ gradient to IPSC decline due to other (likely synaptic) factors. To estimate the latter, we used the recording condition where the HCO₃⁻ current is dominant (at -90mV) because, unlike Cl⁻, the HCO₃⁻ driving force is known to remain stable because it is not rate limited by a transporter (Kaila, *Progress Neurobiol* 1994; Staley *Science* 1995; Hewitt et al., *Nat. Neurosci*, 2009). Thus, the size of the HCO₃⁻ current is not relevant here. Under those recording conditions (-90 mV), the rate of synaptic depression can now be ascribed only to GABA_A desensitization or synaptic fatigue, etc. Subtracting the depression measured at 0mV from that at -90mV allowed us to isolate what was due to Cl⁻ accumulation only. This has now been clarified in the text. Note that former Fig. 4 is now Fig. 6.

4. Figure 5

a. IHC is not a compelling method to demonstrate differences in protein

b. The IHC needs to have a control that corresponds to the total neuronal membrane area in each lamina. Perhaps a microtubule stain?

c. Panel d: the starred p value appears to apply to the last 5 measures only? Is this a measure of probability that the slope is nonzero? Why are only the last 5 points included? How many repeated measures were used – would need to account for all possible subgroups of 5 contiguous points.

We respectfully disagree with the reviewer statement that “IHC is not a compelling method to demonstrate differences in protein”. The reviewer is right however that the data have to be reported by units of membrane. This was perhaps not made clear enough in the previous version, but the quantifications reported in former Fig. 5d and 6b-d were indeed expressed per unit of membrane. Although other methods are available to estimate protein quantity in samples, they hardly allow quantifying protein in microdomains let alone units of membrane area. The combination of high-resolution confocal microscopy with automated or semi-automated quantification tools, on the other hand, can allow for an unbiased analysis of fluorescence intensity per unit of neuronal membrane. Our group has developed a wide array of such quantification approaches (Godin et al *PNAS* 2011, *Biophys J* 2015; Ferrini et al., *Scientific Reports* 2017; Dedek et al., *Brain* 2019; Lorenzo et al., *Nat Commun* 2020). To address the reviewer’s concern, and to strengthen the robustness of our quantification, we now show results of analyses using four different automated and semi-automated unbiased quantification approaches: These are illustrated in new Fig. 4 (former Fig. 5) and supplementary Fig. 4:

1- Fig. 4 b-c, the analysis of KCC2 trans-laminar profile of the dorsal horn KCC2 intensity per pixel was calculated with an automated MATLAB code and related to IB4 staining as a reference for lamina II identification.

2- Fig. 4d The quantification of trans-laminar membrane fluorescence intensity by manually drawing ROIs around randomly selected cell bodies in lamina I and lamina II and plotting the

average fluorescence intensity per unit (pixel) of membrane length against the distance from the superficial dorsal border.

3- Fig. 4e The quantification of membrane KCC2 profile intensity per pixel by using a semi-automated method based on MATLAB code allowing to analysis sub-cellular profile intensity (MASC- π) in identified dorsal horn neurons. Neuronal membranes are firstly delineated by an operator then for each pixel in the region of interest, then the distance to the closest membrane segment was calculated. KCC2 fluorescence signal was quantified by averaging pixel intensities as a function of the distance to the neuron membrane obtaining pixel intensity for membrane unit.

4- Fig. 4f Analysis of KCC2 membrane intensity by global index (MAGI) obtained by automated quantification with MATLAB code which calculates the index of membrane KCC2 intensity after removing intracellular signal from the total signal.

Concerning the reviewer comments regarding the statistical test for previous Fig. 5d (now 4d), we agree that the way statistics and measurements were presented was misleading. We now added a regression analysis indicating that the slope is significantly different from 0. A one-way-RM-ANOVA followed by multiple comparison test confirmed the significant correlation between depth and KCC2 intensity. As per our response to reviewer one, we have now standardized plotting and analysis to the first 80 μm from the white matter border.

5. Figure 7

a. The argument that KCC2 is affecting disinhibition at 2Hz is not congruent with the predictions of figure 3d, which indicates that ECI is stable at that PSC frequency.

b. 2 Hz CI loading is also not congruent with Figure 4a-c where Hz stimuli at 10x higher frequency were used to demonstrate CI loading.

The reference to 2Hz may have been misleading here. There is no relationship between the frequency of the dorsal root afferent stimulation and the frequency at which inhibitory transmission fails in Fig. 6. The relationship between input from afferents and dorsal horn cell output is highly non-linear. i.e. single spikes in single primary afferents generate high frequency bursts of firing in dorsal horn interneurons through polysynaptic pathways (De Koninck & Henry *J. Physiol.* 1994), making the lability of inhibition described in Fig. 6 functionally relevant here. We have clarified this in the text.

As for the protocol we used to induce long term facilitation (former Fig. 7-8; now consolidated into Fig. 7 and New Supplementary Figs. 6-7) it is the standard protocol for the spinal dorsal horn (Ikeda et al, *Science* 2003 and *Science* 2006).

Yet, beyond the facilitation protocol itself, the striking observation we made is the continuous run up of the facilitation, which goes beyond the initial train. This is similar to that highlighted by Ferando et al *Nat. Neurosci.* 2016 (see Supplementary Figs. 3,5,8 of their paper) suggesting that the potentiated post-synaptic response keeps driving the circuit at a level where inhibition is continuously challenged (De Koninck & Henry *PNAS* 1991). We added a statement to this effect in discussion. This is confirmed by our observation that 1) blocking KCC2 replicates this run-up in deeper cells where KCC2 is normally strong (former Fig. 7a-c; now Fig. 7e,g and Supplementary Fig. 6a); 2) blocking BDNF-TrkB signaling (with ANA-12) prevents the run up (former Fig. 7d-f; now Fig. 7f,g and Supplementary Fig. 6b); and we now show in new experiments that 3) a KCC2 enhancer also prevents the run up (new panels Fig. 7g and Supplementary Fig. 7e,f).

c. VU0240551 has substantial nonspecific effects at 10 μ M. These experiments must be repeated with GABA receptors blocked to assess the nonspecific VU024 effects in this preparation.

The concentration of VU0240551 used in our study is extensively published in previous studies to block KCC2 in vitro (of 13 items in *pubmed* in which “VU0240551” is used in vitro or ex-vivo, 9 studies use 10 μ M: Dzhala et al., *J Neurosci* 2012, Hamidi et al., *Pflugers Arch*, 2015; Hamidi and Avoli *Neurobiol Dis.* 2015; Wang et al., *Neuroscience* 2015; Uwera et al., *Brain Res.* 2015, Silvestre de Ferron et al., *Addict Biol.* 2017, Klett and Allen, *Sci. Rep.* 2017, Myers et al., *Epilepsy*, 2018; Balapattabi et al., *J Neuroendocrinol*, 2019).

Nevertheless, to exclude possible off-target effects, and strengthen the link with Cl⁻ transport/loading, we have now added 3 experiments:

1- we repeated the same experiment by substituting VU0240551 with furosemide as in Fig. 1e (Supplementary Fig. 6). In these experiments, in both control and treated conditions, bumetanide was present to subtract any presynaptic NKCC1 contribution (see response to Reviewers 1, point 2).

2- we have used expression of the Cl⁻ pump halorhodopsin (NpHR3.0) to induce a Cl⁻ load by optogenetic activation in dorsal horn neurons and found that it replicated the run up in facilitation (new Supplementary Fig. 7c-d).

3- we have enhanced KCC2 activity by treatment with KCC2 enhancer CLP257 and found that it prevents the run up long term facilitation (new Fig. 7g and Supplementary Fig. 7e,f).

d. Similarly, BDNF TrkB receptors have many effects beyond KCC2 expression, and repeating the experiments with GABA_A receptors blocked are critical.

We agree with the reviewer that blocking TrkB may have effect beyond KCC2. On the other hand, blocking GABA_A receptors also would have very dramatic consequences in the network stability affecting the circuitry at both pre- and postsynaptic sites.

To complement the BDNF-TrkB blockade experiments we thus chose to test the effect of a more direct way to enhance KCC2. As described above, we found that treatment with the KCC2 enhancer CLP257 prevents the run up long term facilitation (new Fig. 7g and Suppl Fig. 7e,f), further supporting the findings obtained with ANA-12 treatment.

Discussion:

Importance of KCC2 vs Donnan and effects of synaptic Cl loading: Please note that the variance in ECl does not decrease when KCC2 is blocked in lamina 1 in Figure 2C. If variance in KCC2 activity accounted for the variance in ECl in Cl-loaded neurons, then blocking KCC2 should have removed this variance. The stable variance in control vs furosemide argues quite strongly for the views expressed in reference 30. So does the variance in ECl shown in figure 2f. This is not to discount the importance of the finding of variance in KCC2 activity, which is another layer of complexity in GABA signaling.

Good observation! We have added a statement to this effect, referring to the Donnan effects presented in Glykys et al *Science* 2014).

Reviewer #3 (Remarks to the Author):

The manuscript by Ferrini et al. describes experiments revealing a difference in distribution of the chloride transporter, KCC2, in the superficial laminae of the dorsal horn. The authors go on to demonstrate that in lamina I (a region that includes key brain-projecting neurons) under conditions of chloride stress and high synaptic activity, KCC2 can no longer keep up with the chloride accumulation and this leads to a functional decrease in inhibition to the region. Lamina II, a heterogeneous group of interneurons, have a higher expression of KCC2 and less effect on inhibition during a chloride load. The authors drive synaptic release of glutamate in vitro using capsaicin and in vivo using dorsal root stimulation, and find that apparent excitatory transmission in lamina I is enhanced considerably more than lamina II. They suggest that their work presents an important new clue to how synaptic activation can promote hyperalgesia by increasing synaptic drive via lamina I neurons.

Overall the work is carefully and well done and the project is of broad interest. However, there is a connection missing between their in vitro work and in vivo work that makes the results less convincing.

We thank the reviewer for its positive comments and for appreciating our work.

Suggestions for revision:

1. The authors have suggested an intriguing interaction between synaptic activation and a failure of inhibition to lamina I neurons. However, there remains an important missing link in the manuscript in its present form. The in vivo field potential recordings are a strength in terms of what may occur under physiological conditions. By contrast, the interpretation is more tenuous. To connect the clean slice recording data of direct measurements of ECl in individual neurons and spontaneous capsaicin-driven IPSCs with the in vivo field potentials during dorsal root stimulation, it would be important to add an experiment in the slice stimulating evoked EPSCs before and after 2Hz stimulation, and preferably repeating the result with the trkB antagonist. Without this experiment, the link between the in vivo pre and post 2Hz stimulation data and the in vitro differences in ECl is missing. Field potentials are inherently complex (see below), and blocking trkB receptors for several hours will have multiple effects on the circuit at many levels other than ECl in the lamina I and II neurons.

We wish to thank the reviewer for the suggestion. However, the proposal by the reviewer to replicate in slices the root stimulation performed on spinal cord explants would not work because the slices miss a lot of the connectivity *in vivo*, which we manage to retain better in the explant. This is illustrated by the fact that input from single afferents generate large bursts of activity in dorsal horn neurons *in vivo* as mentioned above (De Koninck & Henry *J.Physiol.* 1994), yet in slices they merely produce single spikes.

To bolster the parallel between the Cl⁻ homeostasis and synaptic plasticity we instead added more pharmacological investigations linking Cl⁻ homeostasis as a common mechanism responsible for the phenomena observed in slices, spinal cord explants and now (with additional optogenetics experiments) *in vivo*.

Specifically, **1**) we repeated key experiments in spinal cord slices in the presence of TrkB blocker ANA-12, but also using the KCC2 enhancer CLP257 (new Fig. 6d and Supplementary Fig.5a). We also **2**) repeated root stimulation experiments on spinal cord explants in the presence of the KCC2 enhancer CLP257. We found that increasing KCC2 activity also constrained synaptic plasticity in the dorsal horn (new Fig. 7g and Suppl Fig. 7e,f). Finally, **3**) we added a new set of *in vivo* experiments in which a dominant lamina I input (TRPV1- afferents, encoding thermal stimuli) or a dominant lamina II input (MRGPRD- afferents, encoding mechanical stimuli) were optogenetically stimulated. Using this paradigm we showed that uneven strength of inhibition in the SDH deeply impacts in sensitization processes across sensory modalities (greater sensitization in lamina I input). Treatment with a KCC2 enhancer equalized sensitization across modalities. These findings

now more robustly link the plasticity phenomena we observed at each level: from ionic plasticity of synaptic strength to spinal output modulation to behavioural sensitization.

2. Because TrkB inhibition could have multiple effects in the circuit other than direct effects on extrusion of Cl⁻, the TrkB inhibitor ANA should be used to show that four hours after in vivo injection, there is a significant difference in the rundown of eIPSCs during a train in the slice, as shown in figure 4a.

We repeated the experiment in the former Fig 4a (Now Fig.6) in the presence of ANA-12. The results demonstrate that blocking TrkB indeed prevents the collapse of inhibition in lamina I during train of stimulation, making it similar to lamina II. As described above, similar results were obtained by pre-incubation with the KCC2 enhancer CLP257 (Fig. 6d) linking BDNF-TrkB signaling with the collapse in Cl⁻ extrusion.

3. The measurement of the area of a field potential in vivo is misleading. LTP is commonly used to refer to synaptic changes in the monosynaptic EPSC/P, not to the multiple polysynaptic events that follow the initial EPSP in this circuit. While it is difficult to distinguish the two in field potential recordings, the common approach to get around this has been to use field EPSP slope, as this is arguably dominated by earlier events (initial primary afferent EPSPs). Measuring an area over hundreds of milliseconds is not unreasonable for the arguments in this manuscript, including as it does many recurrent synapses, since the loss of inhibition will enhance any EPSC on lamina I cells regardless of whether they originate in the dorsal root or from central neurons. For the main point of this paper, this approach is acceptable as long as the authors make clear that this should not be thought of as LTP in the conventional sense.

We agree that LTP is used to describe a specific monosynaptic phenomenon of synaptic changes. While parallels have been drawn between LTP and synaptic plasticity in the spinal cord (Ji et al. *TINS* 2003, Sandkuhler *J Pain* 2010), field responses in our circuit arise from the activation of multiple components of mono- and postsynaptic events (including slow and fast afferents). Therefore, slope measurements in this circuit may be misleading and we believe that area measurements are more robust.

Despite the link drawn by certain authors with LTP, to be more conservative, we no longer refer to the changes in plasticity as LTP but as long-term facilitation (LTF), which is a term also used to describe this phenomenon in the spinal cord.

4. The “runaway” rise of the field potential area may or may not have anything to do with LTP at the primary afferent synapse, and should be called something else. Moreover, it is notable that the deeper layer field potentials also increase continuously over hours, and that even after treatment with ANA, both deep and superficial field potentials increase over time (albeit at a dramatically shallower slope) is important as well, suggesting that this may be typical of this set of synapses.

As stated above, we no longer refer to this plasticity as LTP but as facilitation, and we have changed the text accordingly. We also used the rate of growth (slope of LTF) of individual experiments to compare the effect of drug treatments in superficial and deep recordings.

Minor:

1. It would be worth pointing out that lamina II neurons (as well as lamina I neurons) are a highly heterogeneous population of cells. Do the immune data suggest that ALL neurons in each lamina show differences in KCC2? A good discussion of this point would be of use to those in the field.

We have added a line in the discussion, commenting on the recent paper by Lee et al., *eLife* 2019. The authors have found differences in inhibitory input between excitatory and inhibitory neurons in lamina II. Their data suggest that excitatory neurons are more sensitive than inhibitory neurons to disinhibition. However, they fail to associate these differences between neuronal types to KCC2. On the other hand, more investigations need to be performed on labeled cell lines to properly address KCC2 heterogeneity.

Reviewers' Comments:

Reviewer #1:

Remarks to the Author:

Ferrini and colleagues have done an excellent job of addressing the critiques raised in the prior review. They have systematically addressed each issue, re-analyzed portions of their data, and performed additional experiments. They have taken what was already a strong paper and made it outstanding. I have just one question/concern that I believe the authors can readily address. A few minor issues with regard to wording were also identified.

Their key observation concerns the gradient in KCC2 activity across lamina I and II, with neurons in lamina I exhibiting greater potential for ionic plasticity. As a result, neurons in lamina I exhibit a runaway facilitation whereas synaptic plasticity in lamina II appears more restrained. As the authors note, these laminae also receive differential input from thermal (lamina I) and mechanical (lamina II) afferents. Given this characterization, one would expect more dramatic amplification of thermal input, as the authors note on lines 383-384. Likewise, they would anticipate that the antinociceptive effect of engaging inhibitory interneurons in the SDH would be weaker for thermal stimuli (372-375). There is, though, some potential for confusion here. What the authors nicely show is that optogenetic activation of TRPV1 primary afferents that project to lamina I induce greater (mechanical) sensitization (relative to MRGPRD afferents that project to lamina II). This supports their general framework, but leaves the reader wondering how the sensitization of fibers in lamina I affects behavioral reactivity to thermal stimulation. I believe that their framework anticipates that noxious input that engages TRPV1 afferents would rapidly sensitize lamina I neurons, producing a thermal hyperalgesia. Yet, unless I missed it, that behavioral effect is not evident in their data. Upon reflection, I wonder if this is an artifact of how many of us routinely test thermal reactivity (using repeated testing). For example, on the thermal tail-flick test, we routinely perform a few (e.g., 3) tests to establish baseline responsiveness, throwing out the first response. We do this because the first response is routinely much longer, whereas subsequent responses typically remain stable over time. Lab lore is that the first response is longer because the tail is cool (due to low blood flow). The shift could, however, reflect the emergence of facilitation within lamina I (initiated by the first encounter with the noxious thermal stimulus). Further, I believe that their framework would predict that pretreatment with capsaicin would eliminate this trial 1 effect. At a functional level, the system might be built this way because organisms often encounter hot objects that can induce tissue damage (e.g., rocks heated by radiant heat). To minimize damage, the system may be built so that the brake on afferent input is quickly removed—after just one encounter.

Minor issues:

On line 31, readers may not understand what “attenuated the KCC2 gradient” means prior to reading the article. Please re-word to clarify.

Line 218: “all these four” could be changed to “all four”.

Line 233: “by using” could be changed to “using”.

Line 238: “from every of the other particles” could be changed to “from every other particle”.

Lines 324-325: I would recommend reminding the reader that ANA-12 is a TrkB antagonist.
Line 391: "should keep into" could be changed to "should take into".
Line 721: "conscious or unconscious" could be deleted.
Line 742: "directly" could be deleted.
Line 749: "on axon" should be "on the axon".
Line 752: "homemade" could be changed to "a custom".
Line 766: "instead" could be deleted.
Line 800: please specify mechanical force of the filament (rather than referring to it as "#5").
Line 843: I would suggest changing "We now describe the simulations whose results are displayed in Fig. 1.g.h." to "The simulation displayed in Fig. 1g.h were derived as follows."
Lines 849 and 873: "Explicitly" could be deleted.

Reviewer #2:

Remarks to the Author:

The manuscript has been substantially improved by careful revision.

One minor concern is a probable typo line 408 - should this be reference 30 vs reference 6?

Reviewer #3:

Remarks to the Author:

The authors have made a good attempt to address the comments of all of the reviewers. However, I continue to find the data regarding BDNF particularly weak. BDNF is in fact a very controversial player in LTP in other circuits and is not widely accepted as part of that phenomenon. Moreover, this extremely potent and widely-acting modulator has multiple effects over several distinct time courses. I do not find the *in vivo* use of the TrkB antagonist convincing, and I think there are multiple alternative explanations of the data that are not included by the authors. I recommend, along with another reviewer, that these data be deleted from the paper or that the conclusions drawn be significantly tempered.

Minor:

I could still wish that the authors would report differences across lamina II in Cl⁻ handling, as there are many different cell types here according to the literature. If ECl is different in many cell types in lamina II, this is quite important; if it is primarily in one cell type, that is also quite important. This is not a criticism, but any light the authors can shed will advance the field.

REVIEWERS' COMMENTS:

Reviewer #1 (Remarks to the Author):

Ferrini and colleagues have done an excellent job of addressing the critiques raised in the prior review. They have systematically addressed each issue, re-analyzed portions of their data, and performed additional experiments. They have taken what was already a strong paper and made it outstanding. I have just one question/concern that I believe the authors can readily address. A few minor issues with regard to wording were also identified.

Their key observation concerns the gradient in KCC2 activity across lamina I and II, with neurons in lamina I exhibiting greater potential for ionic plasticity. As a result, neurons in lamina I exhibit a runaway facilitation whereas synaptic plasticity in lamina II appears more restrained. As the authors note, these laminae also receive differential input from thermal (lamina I) and mechanical (lamina II) afferents. Given this characterization, one would expect more dramatic amplification of thermal input, as the authors note on lines 383-384. Likewise, they would anticipate that the antinociceptive effect of engaging inhibitory interneurons in the SDH would be weaker for thermal stimuli (372-375). There is, though, some potential for confusion here. What the authors nicely show is that optogenetic activation of TRPV1 primary afferents that project to lamina I induce greater (mechanical) sensitization (relative to MRGPRD afferents that project to lamina II). This supports their general framework, but leaves the reader wondering how the sensitization of fibers in lamina I affects behavioral reactivity to thermal stimulation. I believe that their framework anticipates that noxious input that engages TRPV1 afferents would rapidly sensitize lamina I neurons, producing a thermal hyperalgesia. Yet, unless I missed it, that behavioral effect is not evident in their data. Upon reflection, I wonder if this is an artifact of how many of us routinely test thermal reactivity (using repeated testing). For example, on the thermal tail-flick test, we routinely perform a few (e.g., 3) tests to establish baseline responsiveness, throwing out the first response. We do this because the first response is routinely much longer, whereas subsequent responses typically remain stable over time. Lab lore is that the first response is longer because the tail is cool (due to low blood flow). The shift could, however, reflect the emergence of facilitation within lamina I (initiated by the first encounter with the noxious thermal stimulus). Further, I believe that their framework would predict that pretreatment with capsaicin would eliminate this trial 1 effect. At a functional level, the system might be built this way because organisms often encounter hot objects that can induce tissue damage (e.g., rocks heated by radiant heat). To minimize damage, the system may be built so that the brake on afferent input is quickly removed—after just one encounter.

Minor issues:

On line 31, readers may not understand what “attenuated the KCC2 gradient” means prior to reading the article. Please re-word to clarify.

Line 218: “all these four” could be changed to “all four”.

Line 233: “by using” could be changed to “using”.

Line 238: “from every of the other particles” could be changed to “from every other particle”.

Lines 324-325: I would recommend reminding the reader that ANA-12 is a TrkB antagonist.

Line 391: “should keep into” could be changed to “should take into”.

Line 721: “conscious or unconscious” could be deleted.

Line 742: “directly” could be deleted.

Line 749: “on axon” should be “on the axon”.

Line 752: “homemade” could be changed to “a custom”.

Line 766: “instead” could be deleted.

Line 800: please specify mechanical force of the filament (rather than referring to it as “#5”).

Line 843: I would suggest changing “We now describe the simulations whose results are displayed in Fig. 1.g.h.” to “The simulation displayed in Fig. 1g.h were derived as follows.”.

Lines 849 and 873: “Explicitly” could be deleted.

We thank reviewer #1 once again for appreciating our work. All the comments and observations are pertinent and inspiring. The reviewer is right in his observation that engaging “TRPV1 afferents would rapidly sensitize lamina I neurons, producing a thermal hyperalgesia”. In the present study, we have decided to use the traditional von Frey test as behavioral model for testing sensitization for a number of reasons. First, we needed a common scale to properly compare sensitization across sensory modalities; i.e. using the same unit to measure sensitization following both TRPV1- and MRGPRD- stimulation. Testing mechanical threshold by von Frey represented the most logical solution as it provides a quantitative measurement of sensory threshold without inflicting a noxious (and thus, as the reviewer recalled, potentially sensitizing) stimulation. Second, both thermal and mechanical sensitization are known to induce mechanical hypersensitivity, thus alterations in mechanical threshold were expected following stimulation of both MRGPRD and TRPV1 fibers.

On the other hand, the development of thermal hyperalgesia following sensitization across TRPV1 fibers and its relationship with central inhibition needs a dedicated set of experiments. We are currently developing a set of behavioral tools to probe this prediction in both normal and pathological settings.

All the minor points raised by reviewer #1 have been addressed as suggested.

The sentence in line 31, that was unclear, was changed from “attenuated the KCC2 gradient” to “increases KCC2 in lamina I”.

Reviewer #2 (Remarks to the Author):

*The manuscript has been substantially improved by careful revision.
One minor concern is a probable typo line 408 - should this be reference 30 vs reference 6?*

We thank Reviewer #2 for the useful comments at the first submission of this work that helped to improve the manuscript.

The wrong reference has been corrected.

Reviewer #3 (Remarks to the Author):

The authors have made a good attempt to address the comments of all of the reviewers. However, I continue to find the data regarding BDNF particularly weak. BDNF is in fact a very controversial player in LTP in other circuits and is not widely accepted as part of that phenomenon. Moreover, this extremely potent and widely-acting modulator has multiple effects over several distinct time courses. I do not find the in vivo use of the TrkB antagonist convincing, and I think there are multiple alternative explanations of the data that are not included by the authors. I recommend, along with another reviewer, that these data be deleted from the paper or that the conclusions drawn be significantly tempered.

Minor:

I could still wish that the authors would report differences across lamina II in Cl⁻ handling, as there are many different cell types here according to the literature. If ECl⁻ is different in many cell types in lamina II, this is quite important; if it is primarily in one cell type, that is also quite important. This is not a criticism, but any light the authors can shed will advance the field.

We thank Reviewer #3 for appreciating the work done in the revised version of this manuscript.

We agree that BDNF may produce “multiple effects over several distinct time courses”. This was indeed discussed in the revised version of this manuscript in which we highlighted the fact, for

instance, that under certain conditions, or developmental stages, TrkB activation would downregulate KCC2 expression, while in other it would increase it, via the activation of different intracellular pathways. On the other hand, the conclusions we draw are based on the data we obtained either by blocking TrkB with a specific, and now widely used, TrkB antagonist, as well as by analyzing receptor distribution. These results, which are based on quite different approaches, are both consistent with a role of TrkB as negative modulator of KCC2. Moreover, these findings are consistent with recent published data supporting the role of TrkB in promoting LTP-like behavior, at least, in the dorsal spinal cord (Hildebrand et al. Cell rep 2016; Dedek et al., Brain, 2019). Now, to mitigate the emphasis on the role of TrkB signaling, we have removed "TrkB" from the title, and toned down the associated conclusions in the abstract and discussion. Nevertheless, we don't see a specific reason to remove the data from the paper.

We totally agree on the second point raised by the reviewer, i.e. the importance to identify specific cell populations in the dorsal horn according to their Cl^- extrusion capacity. The distribution of E_{GABA} across a broad range of values in both lamina I and II (see Fig. 2b) indicates that Cl^- extrusion capacity is heterogeneous not only between laminae but also within. This encourages the search for a common signature identifying weak Cl^- extruders vs strong Cl^- extruders. We however believe this goes well beyond the scope of the present paper and hope to be able to address this in a future study.